# Replicability in Learning: Geometric Partitions and Sperner-KKM Lemma

**Jason Vander Woude**
Sandia National Laboratories
jasonvwoude@gmail.com

**Peter Dixon**
University of Toronto, Mississauga
tooplark@gmail.com

**A. Pavan**
Iowa State University
pavan@cs.iastate.edu

**Jamie Radcliffe**
University of Nebraska-Lincoln
jamie.radcliffe@unl.edu

**N. V. Vinodchandran**
University of Nebraska-Lincoln
vinod@unl.edu

## Abstract

This paper studies replicability in machine learning tasks from a geometric viewpoint. Recent works have revealed the role of geometric partitions and Sperner's lemma (and its variations) in designing replicable learning algorithms and in establishing impossibility results.

A partition $\mathcal{P}$ of $\mathbb{R}^d$ is called a $(k,\varepsilon)$-secluded partition if for every $\vec{p} \in \mathbb{R}^d$, an $\varepsilon$-radius ball (with respect to the $\ell_\infty$ norm) centered at $\vec{p}$ intersects at most $k$ members of $\mathcal{P}$. In relation to replicable learning, the parameter $k$ is closely related to the *list complexity*, and the parameter $\varepsilon$ is related to the sample complexity of the replicable learner. Construction of secluded partitions with better parameters (small $k$ and large $\varepsilon$) will lead to replicable learning algorithms with small list and sample complexities.

Motivated by this connection, we undertake a comprehensive study of secluded partitions and establish near-optimal relationships between $k$ and $\varepsilon$.

1. We show that for any $(k,\varepsilon)$-secluded partition where each member has at most unit measure, it must be that $k \geq (1 + 2\varepsilon)^d$, and consequently, for the interesting regime $k \in [2^d]$ it must be that $\varepsilon \leq \frac{\log_4(k)}{d}$.

2. To complement this upper bound on $\varepsilon$, we show that for each $d \in \mathbb{N}$ and for each viable $k \in [2^d]$, a construction of a $(k,\varepsilon)$-secluded (unit cube) partition with $\varepsilon \geq \frac{\log_4(k)}{d} \cdot \frac{1}{8\log_4(d+1)}$. This establishes the optimality of $\varepsilon$ within a logarithmic factor.

3. Finally, we adapt our proof techniques to obtain a new "neighborhood" variant of the cubical KKM lemma (or cubical Sperner's lemma): For any coloring of $[0,1]^d$ in which no color is used on opposing faces, it holds for each $\varepsilon \in (0, \frac{1}{2}]$ that there is a point where the open $\varepsilon$-radius $\ell_\infty$-ball intersects at least $(1 + \frac{2}{3}\varepsilon)^d$ colors. While the classical Sperner/KKM lemma guarantees the existence of a point that is "adjacent" to points with $(d+1)$ distinct colors, the neighborhood version guarantees the existence of a small neighborhood with exponentially many points with distinct colors.

## 1 Introduction

Can we design learning algorithms that are replicable? Typically, learning algorithms observe samples from an unknown distribution and output a hypothesis. Since different runs of a learning algorithm

38th Conference on Neural Information Processing Systems (NeurIPS 2024).

may observe different samples, the algorithm may output different hypotheses on different runs, making standard learning algorithms *non-replicable*. Several recent works have been investigating various notions of replicability in learning algorithms. Intuitively, a replicable learning algorithm should output the same canonical hypothesis, on multiple runs (with high probability). However, it has been observed that such an ideal notion of replicability may not be achievable even in simple learning tasks such as learning one-dimensional thresholds. This led to relaxed definitions of replicability including $\rho$-replicability [31], stability [10], global-stability [24], and list-replicability [17].

A significant insight that emerged form these works is the profound connection between geometry and algorithmic replicability [17, 13, 31, 12]. These works use geometric partitions to design replicable algorithms and employ (variants of) Sperner/KKM lemma, including Poincare-Miranda and Borsuk-Ulam theorems, to obtain lower bound results. In particular, in [17], the authors use the notion of *secluded partitions* of $\mathbb{R}^d$ [44] to design replicable learning algorithms with small list complexity. The works of [13, 12, 17] used Sperner/KKM, Poincare-Miranda and Borsuk-Ulam theorems to obtain lower bounds on list complexity and stability parameters for various learning tasks.

Motivated by these connections we undertake an in-depth investigation into *secluded partitions*. Our first contribution is a comprehensive understanding of the optimality of secluded partition constructions. Our second contribution is the discovery of a new neighborhood variant of the Sperner/KKM lemma.

Secluded partitions have found applications beyond replicable learning including deterministic rounding, pseudodeterministic algorithms, and quantum computaion [44, 7]. Moreover, the notion of secluded partitions is simple and natural and is rooted in the works of Lebesgue and Brouwer [36, 8]. Thus, our investigation of secluded partitions should be seen as a fundamental endeavor.

The applicability of the Sperner/KKM lemma, and other equivalent results such as Brouwer's fixed point theorem, the Poincaré-Miranda theorem, and the Lebesgue covering theorem is not just limited to the area of replicable algorithms. These lemmas and their variants have found numerous applications in various contexts: distributed and parallel computing [2, 4, 39, 28, 42, 6], communication complexity [21, 11], computational complexity [25, 35, 40, 16], algorithmic game theory [14, 15], and fair-division [37, 38, 5]. We expect that the neighborhood variant we established will also be useful in computer science applications.

## 2 Preliminaries

A learning algorithm is $k$-*list replicable* if the output of the learning algorithm belongs to a list $\mathcal{L}$ consisting of at most $k$ good hypotheses with high probability. Below is a more formal definition from [17].

Let $\mathcal{X}$ be a domain over which a family of distributions $\mathcal{D}$ are defined, let $\mathcal{H}$ be a set (representing hypotheses), and $err : \mathcal{D} \times \mathcal{H} \to [0, \infty)$ be an error function. A learning algorithm on input $\varepsilon, \delta$ observes $m$ samples from a distribution $D \in \mathcal{D}$ and learns a hypothesis $h \in \mathcal{H}$ with a small error $err(D, h) \leq \varepsilon$.

**Definition 2.1** (List Replicability). *Let $k \in \mathbb{N}$, $\varepsilon \in (0, \infty)$, and $\delta \in [0, 1]$. A learning algorithm $A$ is $(k, \varepsilon, \delta)$-list replicable if there exists $n \in \mathbb{N}$ such that for every $D \in \mathcal{D}$, there exists a list $L \subseteq \mathcal{H}$ of size at most $k$ such that for all $h \in L$, $err(D, h) \leq \varepsilon$, and*

$$\Pr_{s \sim D^n}[A(s, \varepsilon, \delta) \in L] \geq 1 - \delta.$$

*For $k \in \mathbb{N}$, we call $A$ $k$-list replicable if for all $\varepsilon \in (0, \infty)$ and $\delta \in (0, 1]$, $A$ is $(k, \varepsilon, \delta)$-list replicable. We say that $n$ is the* sample complexity *of $A$ and $k$ is the* list complexity *of $A$.*

The above definition captures the ideas over multiple runs of the learning algorithm, we may see at most $k$ different hypotheses. Note that the ideal scenario is when $k = 1$. A generic goal is to design list replicable algorithms with small list and sample complexities. The work of [17] designed list-replicable algorithms for various learning tasks including a general theorem that any concept class that is learnable with $k$ non-adaptive statistical queries has a $k + 1$-list replicable algorithm.

A key ingredient in their list replicable algorithms is the geometric notion of secluded partitions that we define next. Given a point $\vec{p} \in \mathbb{R}^d$, let $\overline{B}_\infty(\varepsilon, \vec{p})$ ($B_\infty^\circ(\varepsilon, \vec{p})$) denote the closed (respectively, open) $\varepsilon$-ball around $\vec{p}$ in the $\ell_\infty$ norm.

**Definition 2.2** ([44]). *A partition $\mathcal{P}$ of $\mathbb{R}^d$ is called a $(k, \varepsilon)$-secluded partition if for every $\vec{p} \in \mathbb{R}^d$, the ball $\overline{B}_\infty(\varepsilon, \vec{p})$ intersects at most $k$ members of $\mathcal{P}$. The parameters $k$ and $\varepsilon$ are called the* degree *and* tolerance *respectively.*

**Remark.** To avoid trivial partitions where each point is a partition member or the entire $\mathbb{R}^d$ is a single partition member, all the partitions considered in this work have non-zero, bounded measure partition members.

It is easy to see that the standard grid partition of $\mathbb{R}^d$ with unit cubes is $(2^d, \frac{1}{2})$-secluded. The following result [44, 30] improves the degree parameter substantially.

**Theorem 2.3.** *There is a $(d + 1, \frac{1}{2d})$-secluded partition where each partition member is a unit cube.*

Sperner/KKM Lemma, Poincaré-Miranda Theorem, and Borsuk-Ulam Theorem can be viewed as *fixed point theorems* and are known to be equivalent to each other (in the sense that any of these theorems can be derived from the other theorem). We state the Sperner/KKM Lemma below. First we introduce the necessary definitions and notation.

Recall that a $d$-dimensional cube is the set $[0, 1]^d$. For a set of colors $C$, a coloring is a mapping $\chi : [0, 1]^d \to C$. For a color $c \in C$, let $X_c$ denote the set of points assigned color $c$ by $\chi$. That is, $X_c = \chi^{-1}(c)$. *A coloring $\chi$ is a Sperner/KKM coloring if no two points from opposite faces of the cube gets the same color.* That is, for every $c \in C$ the set $X_c$ has the property that for each coordinate $i \in [d]$, the projection $\pi_i(X_c) = \{x_i : \vec{x} \in X_c\}$ does not contain both $0$ and $1$.

**Theorem 2.4** (Sperner/KKM). *Given a valid Sperner/KKM coloring of $[0, 1]^d$ by finitely many colors, there exists a point in the closure of at least $d + 1$ different colors.*

# 3 Our Contributions

This section provides an overview of the results established in this work.

**Secluded Partitions and replicability.** The relationship between replicability and partitions can be best abstracted by considering $d$-coin bias estimation problem [17]: given $d$ coins with unknown biases, estimate the biases of each coin within an additive error of $\nu$. The work of [17] showed that a $(k, \varepsilon)$-partition of $\mathbb{R}^d$ yields a $k$-list replicable algorithm for this task. They used a known construction of $(d + 1, \frac{1}{2d})$-secluded partition of $\mathbb{R}^d$ from Theorem 2.3 to obtain a $d + 1$-list replicable algorithm for this task. However, there is a *cost* to replicability. The sample complexity of the replicable algorithm blows up by a factor of $O(d^2)$ (compared to the sample complexity of the non-replicable algorithm). In general a $(k, \varepsilon)$-secluded partition gives rise to a $k$-list replicable algorithm whose sample complexity blows up by a factor of $O(\frac{1}{\varepsilon^2})$ (compared to the "non-replicable" algorithms).

Thus constructions of secluded partitions with low-degree ($k$) and high tolerance ($\varepsilon$), will lead to list replicable algorithms with *low list and sample complexities*. It is also known that the degree parameter $k$ must be at least $d + 1$ [44]. This leads to the following fundamental questions: Can we design secluded partitions that substantially improve the tolerance with little or no degradation of the degree? For example, consider the $(d + 1, \frac{1}{2d})$-secluded partition from Theorem 2.3. Can we improve this and construct a $(d + 1, \omega(\frac{1}{d}))$-secluded partition? Or can we loosen the degree requirement from $k = d + 1$ to $k \in \mathsf{poly}(d)$ in favor of improving the tolerance from $\varepsilon \in \Theta(\frac{1}{d})$ to $\varepsilon \in \omega(\frac{1}{d})$? Our first contribution is the following upper bound result on the tolerance parameter.

**Theorem 3.1.** *Let $d \in \mathbb{N}$, $\varepsilon \in [0, \infty)$, and $\mathcal{P}$ a partition of $\mathbb{R}^d$ such that every member has outer Lebesgue measure at most $1$. Then there exists some $\vec{p} \in \mathbb{R}^d$ such that $B_\infty^\circ(\varepsilon, \vec{p})$ intersects at least $(1 + 2\varepsilon)^d$ members of $\mathcal{P}$. Thus, if $\mathcal{P}$ is a $(k, \varepsilon)$-secluded partition, then $k \geq (1 + 2\varepsilon)^d$. Consequently, if $k \leq 2^d$, then it must be that $\varepsilon \leq \frac{\log_4(k)}{d}$.*

This result shows that even if one relaxes $k \in \mathsf{poly}(d)$, $\varepsilon \in O(\frac{\log d}{d})$. This shows that the $(d + 1, \frac{1}{2d})$-secluded partition construction is near optimal in terms of the tolerance parameter, for degree $d + 1$. Stated in terms of replicability, this result implies one can not hope to design list-replicable algorithms with improved sample complexity using a secluded partitions approach.

Our second result is a construction of secluded partitions for various choices of the degree parameter $k$ with tolerance parameter $\varepsilon$ almost matching the bound from Theorem 3.1. Until this work, we knew

of secluded partitions for only two choices of the degree parameter $k$: the standard grid partition of $\mathbb{R}^d$ that is $(2^d, \frac{1}{2})$-secluded and the $(d+1, \frac{1}{2d})$ partition from Theorem 2.3. We did not know secluded partition constructions for other choices of $k$. Our second main contribution is the construction of near-optimal secluded partitions for all choices of $k$.

**Theorem 3.2.** *Let $d \in \mathbb{N}$ and $k \in [2^d] \setminus [d]$. Then there exists a $(k, \varepsilon)$-secluded unit cube partition of $\mathbb{R}^d$ with $\varepsilon \geq \frac{\log_4(k)}{8d \log_4(d+1)}$.*

Note that by Theorem 3.1, $\varepsilon \leq \frac{\log_4 k}{d}$ and the above construction achieves $\varepsilon \geq \frac{\log_4(k)}{8d \log_4(d+1)}$. Thus this construction is optimal up to $\log$ factors. As a corollary of this construction, we obtain a smooth tradeoff between list and sample complexities for the problem of estimating the bias of $d$-coins

**Corollary 3.3.** *For the $d$-coin bias estimation problem, there exists a $k$-list replicable algorithm with sample complexity $\tilde{O}(\frac{d^2 \log^2 d}{\nu^2 \log^2 k})$, for any $k \in [2^d] \setminus [d]$, per coin.*

For example, if we allow $k = 2^{\sqrt{d}}$, then the sample complexity is $\tilde{O}(\frac{d}{\nu^2})$, per coin.

**Sperner/KKM Lemma and replicability.** While the geometric tool of secluded partitions has been used to design list replicable algorithms, interestingly works that establish lower bounds on the list complexity of replicable algorithms [17, 13, 12] employ geometric/topological tools such as Sperner/KKM Lemma, Poincaré-Miranda Theorem, and Borsuk-Ulam Theorem. For example, the work [17] used Sperner/KKM lemma to establish a lower bound of $d+1$ on the list complexity for the problem of estimating the bias of $d$ coins as well as for the $d$-dimensional threshold learning problem. The work of [13] used Poincaré-Miranda Theorem to establish a lower bound on the list complexity of classes with VC-Dimension $d$. As our third contribution, we generalize the Sperner/KKM Lemma and obtain a neighborhood variant.

**Theorem 3.4** (Neighborhood Sperner/KKM Lemma). *Given a Sperner/KKM coloring of $[0,1]^d$, for any $\varepsilon \in (0, \frac{1}{2}]$, there exists a point $\vec{p} \in [0,1]^d$ such that $B_\infty^\circ(\varepsilon, \vec{p})$ contains at least $\left(1 + \frac{2}{3}\varepsilon\right)^d$ points with distinct colors.*

Sperner/KKM Lemma(Lemma 2.4) states that in valid coloring of the $d$-dimensional hypercube, there is a point $\vec{p}$ whose closure has at least $d+1$ colors. That is, for every $\varepsilon > 0$, the $\varepsilon$-ball around $\vec{p}$ intersects at least $d+1$ colors. Our theorem is a quantitative generalization of this result. It states that the $\varepsilon$-ball around $\vec{p}$ intersects exponentially many colors—at least $(1 + \frac{2}{3}\varepsilon)^d$ many colors.

## 4 Related Work

One of the first works that studied replicability in the context of learning algorithms is the seminal work of Bun, Livny, and Moran [10]; they used the term *global stability* to capture this notion. A learning algorithm $A$ to be $(n, \eta)$-*globally stable* with respect to a distribution $D$ if there is a hypothesis $h$ such that $\Pr_{S \sim D^n}(A(S) = h) \geq \eta$, here $\eta$ is called the *stability parameter*. They used the notion of stability, combined with the work of [3], to obtain the equivalence between online learnability and differentially private PAC learnability. Ghazi, Kumar and Manurangsi [24] generalized the notion of stability to pseudo-global stability and list-global stability. Impagliazzo, Lei, Pitassi, and Sorrell [31] introduced the notion of $\rho$-replicabililty and designed replicable algorithms for various learning tasks. One of their replicable algorithms uses a partition/tiling known as "foams tiling" [34]. Dixon, Pavan, Vander Woude and Vinodchandran [17] studied the notions of *list replicability* and *certificate replicability* as a measure of the degree of (non)-replicability. Chase, Moran and Yehudayof [13] related the notions list complexity and stability. They established that a learning task has list complexity $k$ if and only if its stability parameter is $1/k$. They also established lower bounds the list-complexity (upper bound on the stability) on the PAC-learnability of classes with bounded VC-dimension. In [12], the authors use Borsuk-Ulam theorem to establish impossibility results for replicable agnostic PAC learning.

In the context of randomized algorithms, the notion of replicability is studied under the terminology *pseudodeterminism*. This notion was introduced by Gat and Goldwasser [23] and has been extended to notions called multi-pseudodeterminism [26] and influential-bit algorithms [27]. The study of replicability in the context of learning has been receiving growing attention over the past few years

and researchers have been investigating this notion under various scenarios. The notion of replicability in the context of stochastic bandits and reinforcement learning has been studied in [19, 33, 18], and the work of [1] studies replicability for optimization problems. Other very recent works include those reported in [9, 29, 32]. A notion that is related to replicability is that of reproducibility. The article by [41] distinguishes these two notions.

# 5 Proof Sketches of Main Results

The main technical tools that we use come from measure theory and the geometry of numbers and include generalized Brunn-Minkowski Inequality and ideas from the standard proof of Blichfeldt's theorem. The generalized Brunn-Minkowski inequality gives a lower bound on the measure of a Minkowski sum of sets ($A + B = \left\{\vec{a} + \vec{b} \colon \vec{a} \in A, \vec{b} \in B\right\}$) based on the measures of those sets. We use the following version of the statement from [22, Equation 11]. Here $m(A)$ is the Lebesgue measure of a set $A \subseteq \mathbb{R}^d$.

**Theorem 5.1** (Generalized Brunn-Minkowski Inequality). *Let $d \in \mathbb{N}$ and $A, B \subseteq \mathbb{R}^d$ be Lebesgue measurable such that $A + B$ is also Lebesgue measurable. Then*

$$m(A + B) \geq \left[m(A)^{\frac{1}{d}} + m(B)^{\frac{1}{d}}\right]^d.$$

A common technique in the proof of Blichfeldt's theorem is to use an averaging argument to show that if a set $A$ is covered by a large family of other sets, then some point in $A$ is covered many times.

## 5.1 Proof Sketch of Theorem 3.1

We present the high-level ideas and the intuition behind the proof of Theorem 3.1. In the appendix, we provide a complete proof. Figure 1 serves as a visual.

The goal is to find some point $\vec{p} \in \mathbb{R}^d$ such that $B_\infty^\circ(\varepsilon, \vec{p})$ intersects at least $(1 + 2\varepsilon)^d$ members of the partition. Instead of directly trying to establish this, we take a critical change of perspective: for any $\vec{p} \in \mathbb{R}^d$ and $X \in \mathcal{P}$ (or really any $X \subseteq \mathbb{R}^d$), it holds that $B_\infty^\circ(\varepsilon, \vec{p}) \cap X \neq \emptyset$ if and only if $\vec{p} \in \bigcup_{\vec{x} \in X} B_\infty^\circ(\varepsilon, \vec{x})$. Thus, what we do is to "replace" every member $X$ of the partition with the enlarged set $\bigcup_{\vec{x} \in X} B_\infty^\circ(\varepsilon, \vec{x})$ and try to find a point $\vec{p}$ that belongs to at least $(1 + 2\varepsilon)^d$ of these enlarged sets. To achieve this, we take inspiration from a common proof of Blichfeldt's theorem—specifically, the following result which says that if we have a collection of sets $A_1, A_2, A_3, \ldots$ which are subsets of another set $S$, then there is a point in $S$ occurring in multiple $A_i$s provided together the $A_i$s have enough volume/measure. We can in fact give a lower bound on the number of $A_i$s to which such a point belongs to. The following is the formal claim of this known result.

**Proposition 5.1** (Continuous Multi-Pigeonhole Principle). *Let $d \in \mathbb{N}$ and $S \subset \mathbb{R}^d$ be bounded and measurable. Let $\mathcal{A}$ be a family of measurable subsets of $S$, and let $k = \left\lceil \frac{\sum_{A \in \mathcal{A}} m(A)}{m(S)} \right\rceil$. Then if $k < \infty$, there exists $\vec{p} \in S$ such that $\vec{p}$ belongs to at least $k$ members of $\mathcal{A}$. (And if $k = \infty$, then for any $n \in \mathbb{N}$ there exists $\vec{p}^{(n)} \in S$ such that $\vec{p}^{(n)}$ belongs to at least $n$ members of $\mathcal{A}$.)*

There is an immediate issue we have to deal with to be able to use Proposition 5.1 for our application. We would like to take $\mathcal{A}$ to be the indexed collection of enlarged partition members: $\mathcal{A} = \left\{\bigcup_{\vec{x} \in X} B_\infty^\circ(\varepsilon, \vec{x})\right\}_{X \in \mathcal{P}}$, but then all we know is that each of these is a subset of $S = \mathbb{R}^d$ which is not bounded. This is a simple enough issue to deal with using a standard measure theory technique of considering instead a sequence $S_1, S_2, S_3, \ldots$ of sets which *are* bounded and get larger and larger so that $\bigcup_{n=1}^\infty S_n = \mathbb{R}^d$; we work with each of these sets individually and then try to use a limiting argument to pass the result back to $S = \mathbb{R}^d$. Specifically, we will take $S_n = [-n, n]^d$ as illustrated in the first two panes of Figure 1. The third pane of Figure 1 illustrates that we will specifically consider the partition of $S_n$ induced by $\mathcal{P}$ which we denote by $\mathcal{S}_n$. That is, the induced partition $\mathcal{S}_n$ is the set $\{X \cap S_n \colon X \in \mathcal{P} \text{ and } X \cap S_n \neq \emptyset\}$. Then for each $S_n$ we consider a collection $\mathcal{A}_n$ of the enlarged members of the induced partition: $\mathcal{A}_n = \{A_Y\}_{Y \in \mathcal{S}_n}$ where $A_Y \overset{\text{def}}{=} \bigcup_{\vec{y} \in Y} B_\infty^\circ(\varepsilon, \vec{y})$. Note that each $A_Y$ is a subset of $S_n' \overset{\text{def}}{=} [-(n + \varepsilon), n + \varepsilon]^d$ as in the fourth pane of Figure 1.

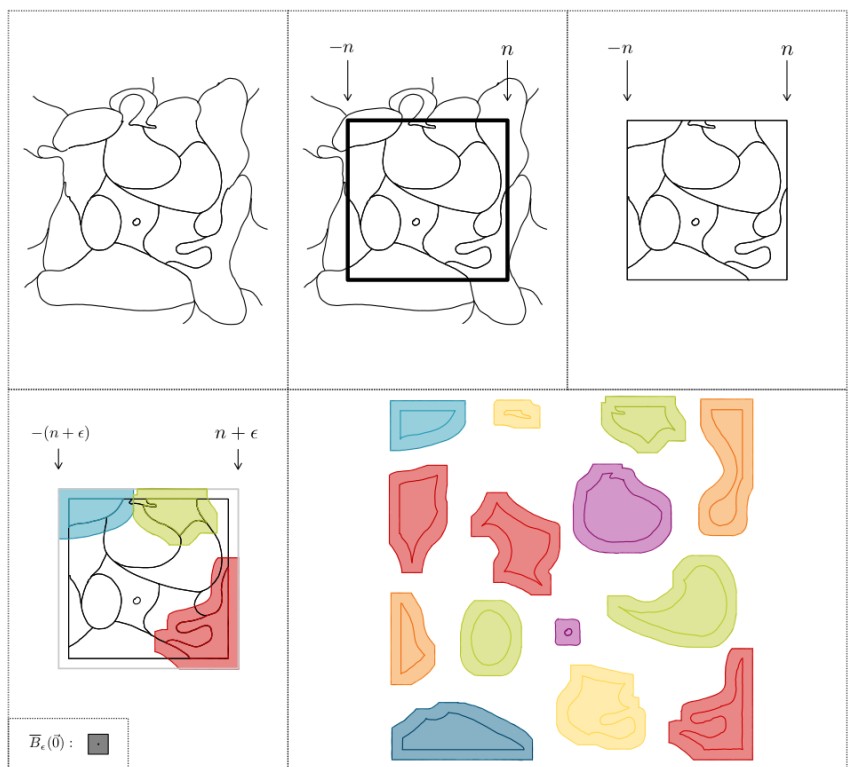

Figure 1: Pane 1: A partition of $\mathbb{R}^2$. Pane 2: We consider only members of the partition which intersect $[-n, n]^2$. Pane 3: The partition that $\mathcal{P}$ induces on $[-n, n]^2$. Pane 4: We enlarge each member by placing an $\varepsilon$-ball at each point of the member. These enlarged elements are still contained within $[-(n+\varepsilon), n+\varepsilon]^2$. Pane 5: The sum of areas of the expanded members is "significantly" more than the area of $[-(n+\varepsilon), n+\varepsilon]^2$.

However, there remains one other issue to deal with to utilize Proposition 5.1—for each $n$, we have to have some lower bound on the expression $\left\lceil \frac{\sum_{A_Y \in \mathcal{A}_n} m(A_Y)}{m(S'_n)} \right\rceil$. We know that $m(S'_n) = (2(n+\varepsilon))^d$, and using the fact that $\mathcal{S}_n$ is a partition of $S_n = [-n, n]^d$, we have the following if $\mathcal{S}_n$ is countable[1] (meaning finite or countably infinite):

$$\sum_{A_Y \in \mathcal{A}_n} m(A_Y) \geq \sum_{Y \in \mathcal{S}_n} m(Y) = m\left( \bigsqcup_{Y \in \mathcal{S}_n} Y \right) = m(S_n),$$

but this is not nearly good enough, because it just gives

$$\left\lceil \frac{\sum_{A_Y \in \mathcal{A}_n} m(A)}{m(S'_n)} \right\rceil \geq \left\lceil \frac{m(S_n)}{m(S'_n)} \right\rceil \geq \left\lceil \frac{(2n)^d}{(2(n+\varepsilon))^d} \right\rceil = \left\lceil \left( \frac{n}{n+\varepsilon} \right)^d \right\rceil = 1$$

whereas we want it $\geq (1 + 2\varepsilon)^d$. Basically, this lower bound is not good because we did not account for the fact that each $A_Y \in \mathcal{A}_n$ is enlarged from $Y \in \mathcal{S}_n$. Thus, we want some way to give for each $Y \in \mathcal{S}_n$, a lower bound on the measure of the enlarged set $A_Y$. One might observe that enlarging with an $\varepsilon$-ball looks something like scaling by a factor of $1 + \varepsilon$, and since the Lebesgue measure (i.e. typical notion of volume/measure in $\mathbb{R}^d$) has the property that scaling by $(1 + \varepsilon)$ increases the measure by a factor of $(1 + \varepsilon)^d$, we might be able to show that the enlarged version of each member increases by a factor of $(1 + \varepsilon)^d$ (which is basically what we are looking to get).

---

[1]If $\mathcal{S}_n$ is uncountable, one of the equalities in the chain becomes the wrong inequality as we get that $\sum_{Y \in \mathcal{S}_n} m(Y) \leq m\left( \bigsqcup_{Y \in \mathcal{S}_n} Y \right)$ by B.2 in B.

This intuition holds, though the actual reason is not related to scaling, and is dependent on the members having measure at most 1. Rather, we use a specialized adaption of Theorem 5.1 to show that

$$m(A_Y) \geq m(Y) \cdot (1 + 2\varepsilon)^d \qquad (1)$$

holds. Now that we have dealt with both issues that arise with trying to apply Proposition 5.1, we can consider a fixed $n$ and can continue. We proceed in two cases: (1) the interesting case in which $\mathcal{S}_n$ has only countably many members, and (2) the nearly trivial case in which the partition $\mathcal{S}_n$ contains uncountably many members. In case (1) we have

$$\left\lceil \frac{\sum_{A_Y \in \mathcal{A}_n} m(A_Y)}{m(S'_n)} \right\rceil = \left\lceil \frac{\sum_{Y \in \mathcal{S}_n} m(A_Y)}{m(S'_n)} \right\rceil \qquad \text{(Re-index)}$$

$$\geq \left\lceil \frac{\sum_{Y \in \mathcal{S}_n} \left[ m(Y) \cdot (1 + 2\varepsilon)^d \right]}{m(S'_n)} \right\rceil \qquad \text{( Equation 1)}$$

$$= \left\lceil \frac{(1 + 2\varepsilon)^d \cdot \sum_{Y \in \mathcal{S}_n} m(Y)}{m(S'_n)} \right\rceil \qquad \text{(Linearity of summation)}$$

$$= \left\lceil \frac{(1 + 2\varepsilon)^d \cdot m\left( \bigsqcup_{Y \in \mathcal{S}_n} Y \right)}{m(S'_n)} \right\rceil \qquad \text{(Countable additivity of measures)}$$

$$= \left\lceil \frac{(1 + 2\varepsilon)^d \cdot m(S_n)}{m(S'_n)} \right\rceil \qquad (S_n = \bigsqcup_{Y \in \mathcal{S}_n} Y)$$

$$= \left\lceil (1 + 2\varepsilon)^d \cdot \left( \frac{n}{n + \varepsilon} \right)^d \right\rceil \qquad (\frac{m(S_n)}{m(S'_n)} = (\frac{n}{n+\varepsilon})^d \text{ as above})$$

Since (for fixed $d$ and $\varepsilon$) we have $\lim_{n \to \infty} \left( \frac{n}{n+\varepsilon} \right)^d = 1$, then $\lim_{n \to \infty} (1 + 2\varepsilon)^d \cdot \left( \frac{n}{n+\varepsilon} \right)^d = (1 + 2\varepsilon)^d$, so because there is a ceiling involved, we can take $N \in \mathbb{N}$ to be large enough that $\left\lceil (1 + 2\varepsilon)^d \cdot \left( \frac{N}{N+\varepsilon} \right)^d \right\rceil = \left\lceil (1 + 2\varepsilon)^d \right\rceil$ , so by Proposition 5.1, there is a point $\vec{p} \in S'_n$ that is contained in at least $(1 + 2\varepsilon)^d$ many sets in $\mathcal{A}_N$, and by our change of perspective, this point $\vec{p}$ has the property that $B^\circ_\infty(\varepsilon, \vec{p})$ intersects at least $(1 + 2\varepsilon)^d$ many members of $\mathcal{P}$.

In case (2) where some $\mathcal{S}_N$ contains uncountably many members, then we completely ignore the lower bound for $m(A_Y)$ in Equation 1 because it might be that lots of members $Y$ (possibly all of them) have measure 0, and so that bound only tells us that $m(A_Y) \geq 0$. Instead, we note that $Y$ is at least non-empty, so contains at least one point $\vec{y}$, and thus $A_Y \supseteq B^\circ_\infty(\varepsilon, \vec{y}) = \prod_{i=1}^d [y_i - \varepsilon, y_i + \varepsilon]$, and so $m(A_Y) \geq (2\varepsilon)^d$. Thus, $\left\lceil \frac{\sum_{A_Y \in \mathcal{A}_N} m(A_Y)}{m(S'_N)} \right\rceil = \infty$, so by Proposition 5.1, there is a point $\vec{p} \in S'_N$ that is contained in at least $(1 + 2\varepsilon)^d$ many sets in $\mathcal{A}_N$, and by our change of perspective, this point $\vec{p}$ has the property that $B^\circ_\infty(\varepsilon, \vec{p})$ intersects at least $(1 + 2\varepsilon)^d$ many members of $\mathcal{P}$.

## 5.2 Proof of Theorem 3.2

The partitions that we construct, to establish Theorem 3.2, are of a very natural form: we build a partition of a large dimension $d$ space, by splitting up the coordinates into smaller sets, and separately partitioning each set of coordinates. In the end, the smaller partitions will be known partition constructions from Theorem 2.3. We will define the construction very generically. We need the observation that if a partition is $(k, \varepsilon)$-secluded, then we can *increase* $k$ to $k'$ and *decrease* $\varepsilon$ to $\varepsilon'$ and the partition is trivially $(k', \varepsilon')$-secluded. We refer to this property as the "monotonicity" property.

**Definition 5.2** (Partition Product). *Let $d_1, \ldots, d_n \in \mathbb{N}$ and $\mathcal{P}_1, \ldots, \mathcal{P}_n$ be partitions of $\mathbb{R}^{d_1}, \ldots, \mathbb{R}^{d_n}$ respectively. Letting $d = \sum_{i=1}^n d_n$, we define the product partition of $\mathbb{R}^d$ as*

$$\prod_{i=1}^n \mathcal{P}_i \overset{\text{def}}{=} \left\{ \prod_{i=1}^n X_i \colon X_i \in \mathcal{P}_i \right\}$$

*where $\prod_{i=1}^{n} X_i$ is viewed as a subset of $\mathbb{R}^d$.*

We specifically stated that $\prod_{i=1}^{n} X_i$ is viewed as a subset of $\mathbb{R}^d$, because technically it is a subset of $\prod_{i=1}^{n} \mathbb{R}^{d_i}$, but this is naturally isomorphic to $\mathbb{R}^d = \mathbb{R}^{\sum_{i=1}^{n} d_i}$. For example, technically, if $d_1 = d_2 = d_3 = 2$, then the elements of $\prod_{i=1}^{n} \mathbb{R}^{d_i}$ are of the form $\langle\langle x_1, x_2\rangle, \langle x_3, x_4\rangle, \langle x_5, x_6\rangle\rangle$, but this is trivially isomorphic to $\mathbb{R}^6$ by instead considering the element as $\langle x_1, x_2, x_3, x_4, x_5, x_6\rangle$.

We can now show that if we take a product of partitions, and we have a guarantee for each $\mathcal{P}_i$ that it is $(k_i, \varepsilon_i)$-secluded, then we can guarantee the product partition is $(k, \varepsilon)$-secluded where $k$ is the product of the $k_i$'s and $\varepsilon$ is the minimum of the $\varepsilon_i$'s.

**Proposition 5.2** (Product Partition Seclusion Guarantees). *Let $n \in \mathbb{N}$. For each index $i \in [n]$, let $d_i, k_i \in \mathbb{N}$, $\varepsilon_i \in (0, \infty)$ and $\mathcal{P}_i$ be a $(k_i, \varepsilon_i)$-secluded partition of $\mathbb{R}^{d_i}$. Then the product partition $\mathcal{P} = \prod_{i=1}^{n} \mathcal{P}_i$ is a $(k, \varepsilon)$-secluded partition of $\mathbb{R}^d$ where $d = \sum_{i=1}^{n} d_i$, and $k = \prod_{i=1}^{n} k_i$, and $\varepsilon = \min_{i \in [n]} \varepsilon_i$.*

*Proof Sketch.* The basic idea is that for any point $\vec{p} \in \mathbb{R}^d$, we consider how many members of $\mathcal{P}$ intersect $\overline{B}_\varepsilon(\vec{p})$. Conceptually we think of $\vec{p}$ as a sequence $\langle \vec{p}^{(i)} \rangle_{i=1}^{n}$ of $n$ points where the $i$th point $\vec{p}^{(i)}$ belongs to $\mathbb{R}^{d_i}$. Because we are working with the $\ell_\infty$ norm (that is the norm used by the definition of secluded), the $\varepsilon$ ball around $\vec{p}$ is the product of the $\varepsilon$ balls around each $\vec{p}^{(i)}$ which is smaller than the product of $\varepsilon_i$ balls around each $\vec{p}^{(i)}$ because we chose $\varepsilon$ as the minimum size. Thus, if the $\varepsilon$ ball around $\vec{p}$ intersects a member $X$ of the partition $\mathcal{P}$, then conceptually viewing $X$ as a sequence $\langle X_i \rangle_{i=1}^{n}$ where $X_i$ is a member of $\mathcal{P}_i$, it must be for each $i \in [n]$ that the $\varepsilon$ ball around $\vec{p}^{(i)}$ intersects $X_i$ (and thus so does the $\varepsilon_i$ ball since $\varepsilon_i \geq \varepsilon$). This means (for each $i \in [n]$) that $X_i$ is one of at most $k_i$ members of $\mathcal{P}_i$ because at most $k_i$ members of $\mathcal{P}_i$ intersect the $\varepsilon_i$ ball around $\vec{p}^{(i)}$ (by definition of $\mathcal{P}_i$ being $(k_i, \varepsilon_i)$-secluded). Thus $X$ is one of at most $\prod_{i=1}^{n} k_i = k$ members of $\mathcal{P}$. That is, there are at most $k$ members of $\mathcal{P}$ that intersect the $\varepsilon$ ball around $\vec{p}$ which is the definition of $\mathcal{P}$ being $(k, \varepsilon)$-secluded. $\square$

Utilizing the construction above, we will now take the unit cube partition from Theorem 2.3 for each $\mathbb{R}^{d_i}$ and take their product to obtain a new partition. Since the dimension of each $d_i$ is smaller than the dimension $d$, this allows us to get a larger value of $\varepsilon_i$ for each partition, and thus a larger value of $\varepsilon$ for the partition of $\mathbb{R}^d$ than if we had used one of the original partitions. The price we pay for this is that the value of $k$ also increases.

We establish the following lemma.

**Lemma 5.3.** *Let $d \in \mathbb{N}$ and $d' \in [d]$. There exists a $(k, \varepsilon)$-secluded unit cube partition of $\mathbb{R}^d$ where $k = (d' + 1)^{\lceil \frac{d}{d'} \rceil}$ and $\varepsilon = \frac{1}{2d'}$.*

*Proof.* Let $n = \lceil \frac{d}{d'} \rceil$. By [44], let $\mathcal{P}'$ be a $(d' + 1, 1/2d')$-secluded unit cube partition of $\mathbb{R}^{d'}$. By Proposition 5.2 $\mathcal{P} = \prod_{i=1}^{n} \mathcal{P}'$ is a $(k, \varepsilon)$-secluded unit cube partition of $\mathbb{R}^{n \cdot d'}$ where $k = (d' + 1)^n$ and $\varepsilon = \frac{1}{2d'}$. Since $n \cdot d' = \lceil \frac{d}{d'} \rceil \cdot d' \geq d$, this trivially (by ignoring extra coordinates) gives a partition of $\mathbb{R}^d$ with these same properties which completes the proof. $\square$

Finally, we are ready to prove Theorem 3.2.

*Proof.* Let $d' \in [d]$ be the minimum integer such that $(d' + 1)^{\lceil \frac{d}{d'} \rceil} \leq k$ and let $\varepsilon = \frac{1}{2d'}$. By Lemma 5.3 and monotonicity, let $\mathcal{P}$ be a $(k, \varepsilon)$-secluded unit cube partition of $\mathbb{R}^d$. Now we prove the bound on $\varepsilon$ in two cases: either $d' = 1$ or $d' \geq 2$.

In the case that $d' = 1$, then $\varepsilon = \frac{1}{2}$ and $k \geq (d' + 1)^{\lceil \frac{d}{d'} \rceil} = 2^d$ and by hypothesis $k \leq 2^d$, so we have equality and we conclude that

$$\frac{\log_4(k)}{4d \log_4(d+1)} = \frac{d/2}{4d \log_4(d+1)} = \frac{1}{8 \log_4(d+1)} \leq \frac{1}{8 \cdot \frac{1}{2}} \leq \frac{1}{2} = \varepsilon$$

which proves the bound on $\varepsilon$ in the first case.

In the other case, we have $d \geq d' \geq 2$. Let $d'' = d' - 1 > 0$ and $\delta = \frac{1}{2d''}$. Note that

$$\frac{\varepsilon}{\delta} = \frac{2d''}{2d'} = \frac{d'-1}{d'} = 1 - \frac{1}{d'} \geq 1 - \frac{1}{2} = \frac{1}{2}$$

so $\varepsilon \geq \frac{1}{2}\delta$. By our choice of $d'$ and because $d'' < d'$, it must be that $k \leq (d'' + 1)^{\lceil \frac{d}{d''} \rceil}$. Noting that $\lceil \frac{d}{d''} \rceil \leq \frac{d}{d''} + 1 = \frac{d+d''}{d''} \leq \frac{2d}{d''}$, we have

$$k \leq (d'' + 1)^{\lceil \frac{d}{d''} \rceil} \leq (d'' + 1)^{\frac{2d}{d''}} = (d'' + 1)^{4d\delta} \leq (d'' + 1)^{8d\varepsilon} \leq (d + 1)^{8d\varepsilon}$$

By taking the logarithm of each side and solving for $\varepsilon$, we obtain the desired result that $\varepsilon \geq \frac{\log_4(k)}{8d \log_4(d+1)}$.

$\square$

## 5.3 Proof Discussion of Theorem 3.4

Interestingly, the proof of the neighborhood Sperner/KKM lemma relies on the techniques developed to prove Theorem 3.1: for each color $c \in C$, let $X_c$ be the set of points that are colored $c$. We union an $\varepsilon$-ball at each point in $X_c$, to obtain an enlarged version $A_{X_c}$ of $X_c$. Now, as before, by the Continuous Pigeonhole Principle (Corollary 5.1 ), there is a point that belongs to many of the enlarged sets (and so the ball located at that point intersects many of the original color sets). However, there are some additional issues that arise on the unit cube that don't arise in $\mathbb{R}^d$.

In the discussion in the proof sketch of Theorem 3.1, the enlarged set was not contained in the original region (denoted $S_n$) and we needed to consider a larger region (denote $S'_n$) to contain them. In $\mathbb{R}^d$ we could deal with this via a limiting argument so that the ratio of the volume change $m(S_n)/m(S'_n)$ tends to 1 (i.e. it became negligible when ceilings were involved). If one enlarges every color in a unit cube $[-\frac{1}{2}, \frac{1}{2}]^d$ in the same way, the measure of each color is guaranteed to increase by a factor of $(1 + 2\varepsilon)^d$ as before, but also the smallest set that contains all of these enlargements is the unit cube $[-\frac{1}{2} - \varepsilon, \frac{1}{2} + \varepsilon]^d$ which increased in measure by a factor of $(1 + 2\varepsilon)^d$ compared to the original cube, so nothing has been gained! Obviously, there will be an overlap of the enlargements, but the bounds given by the generalized Brunn-Minkowsi inequality tell us no additional information.

We resolve this by employing a trick of first extending the coloring directly to $[-\frac{1}{2} - \varepsilon, \frac{1}{2} + \varepsilon]^d$ in a natural way that ensures each color is bounded away from the boundary so that we can perform an enlargement which is non-uniform (it enlarges only toward one of the $2^d$ orthants) and still have the enlarged color set contained in $[-\frac{1}{2} - \varepsilon, \frac{1}{2} + \varepsilon]^d$. This means we end up knowing that each modified color has increased in measure by at least a factor of $(1 + \frac{2}{3}\varepsilon)^d$ and that the modified containing region has not changed in measure at all.

## 6 Conclusion and Open Questions

This work is a comprehensive study of secluded partitions. Prior to this work, it was known that the $(d + 1, O(\frac{1}{d}))$-secluded partitions of [30, 44] were optimal in regards to the degree parameter ($k = d + 1$). However, it was unknown if they were optimal in the tolerance parameter $\varepsilon$ for this choice of $k$. The present work showed that these constructions are optimal in $\varepsilon$ up to a logarithmic factors. Furthermore, they remain optimal within a logarithmic factor even if we allow the degree $k$ to be polynomial in the dimension $d$. We also constructed secluded partitions, optimal up to logarithmic factors, for all $k$ between $d + 1$ and $2^d$.

This work raises a few open problems. At first glance, it might seem to complete the study of secluded partitions; however, it only establishes near-matching bounds for the $\ell_\infty$ norm. In Appendix A.8, we present upper bounds on $\varepsilon$ in terms of $k$ for $\ell_p$ norms, but no known constructions approach these bounds. Developing near-optimal partitions for other norms and exploring their applications to replicability would be an intriguing direction for future research

In replicable algorithm design, there appears to be a *cost* to achieve replicability—sample complexity blow-up. This work showed that this blow-up in sample complexity is unavoidable in list replicability if one uses secluded partitions method. Can we establish a generic lower bound on the sample complexity of list replicable learning algorithms?

While Theorem 3.4 gives a new "neighborhood" variant of the Sperner/KKM lemma, the color bound of $(1 + \frac{2}{3}\varepsilon)^d$ is not tight for small $\varepsilon$ because the standard cubical Sperner/KKM lemma (2.4) shows that even for arbitrarily small $\varepsilon$, the color bound is at least $d + 1$. A compelling research goal is to improve on Theorem 3.4 so that the standard cubical Sperner/KKM lemma follows as a special case. Finally, finding applications of the neighborhood Sperner/KKM lemma is an interesting research direction.

## 7   Acknowledgements

We thank the anonymous reviewers for their valuable suggestions, which improved the presentation of this paper. Vinodchandran's work is partly supported by NSF grants 2130608, 2342244, and a UNL Grand Challenges Grant. Pavan's work is supported in part by NSF grants 2130536 and 2342245. Jason Vander Woude's contributions were made during his time at University of Nebraska, Lincoln and was partly supported by NSF grant 2130608.

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

# A Complete Proofs

## A.1 Notation

The following is a list of some of the notation we will use in this paper.

- We use $\mathbb{N}$ to denote the natural numbers starting with 1.

- We continue to use $\overline{B}_{\|\cdot\|}(\varepsilon, \vec{p})$, $B_{\|\cdot\|}^{\circ}(\varepsilon, \vec{p})$, $\overline{B}_{\infty}(\varepsilon, \vec{p})$, and $B_{\infty}^{\circ}(\varepsilon, \vec{p})$ as before.

- For two sets $A, B \subseteq \mathbb{R}^d$ we write $A + B$ to represent the Minkowski sum $A + B \stackrel{\text{def}}{=} \left\{ \vec{a} + \vec{b} \colon \vec{a} \in A,\ \vec{b} \in B \right\}$. We also may write $\vec{a} + B$ to mean $\left\{ \vec{a} + \vec{b} \colon \vec{b} \in B \right\}$ for some fixed vector $\vec{a}$.

- We will use $v_{\|\cdot\|, d}$ to represent the Borel/Lebesgue measure of the unit radius ball in $\mathbb{R}^d$ with respect to a general norm $\|\cdot\|$. This is a normalization factor that appears in some results.

## A.2 Proof Theorem 3.1

In this section, we present a complete proof of 3.1. We begin with some prerequisite results in A.2.1. Then, in A.2.2 we establish a more generic theorem that can handle any norm. Theorem 3.1 follows as immediate corollary this generic result.

### A.2.1 Prerequisite Results

In this section, we will deal with arbitrary norms of $\mathbb{R}^d$. We point out the well-known fact that all norms on $\mathbb{R}^d$ are equivalent in the sense that they all generate the same topology on $\mathbb{R}^d$. Given two norms $\|\cdot\|^a$ and $\|\cdot\|^b$ on $\mathbb{R}^d$, there exists fixed constants $c_d, C_d \in (0, \infty)$ such that for all vectors $\vec{x} \in \mathbb{R}^d$, it holds that $c_d \|\vec{x}\|^a \leq \|\vec{x}\|^b \leq C_d \|\vec{x}\|^a$. Thus the collection of open sets in $\mathbb{R}^d$ is the same no matter which norm we are using. This also means that the Borel and Lebesgue $\sigma$-algebras on $\mathbb{R}^d$ are the same no matter which norm is used, and thus balls with respect to any norm on $\mathbb{R}^d$ are measurable.

We begin with four simple facts. The first fact will later allow us to pass a result through a limit since the answer will be an integer.

**Fact A.1.** *For any $\alpha \in \mathbb{R}$, there exists $\gamma \in \mathbb{R}$ such that $\gamma < \alpha$ and $\lceil \gamma \rceil = \lceil \alpha \rceil$.*

The next fact says that the Minkowski sum of a set $X$ and an open ball at the origin can be viewed as a union of open balls positioned at each point of $X$.

**Fact A.2.** *For any normed vector space, given a set $X$ and $\varepsilon \in [0, \infty)$, then*

$$X + B_{\|\cdot\|}^{\circ}(\varepsilon, \vec{0}) = \bigcup_{\vec{x} \in X} B_{\|\cdot\|}^{\circ}(\varepsilon, \vec{x}).$$

*The same can be said replacing open balls with closed balls.*

The third fact says that we can decompose a ball into a Minkowski sum of two smaller balls.

**Fact A.3.** *For any normed vector space, and any $\alpha, \beta \in (0, \infty)$, it holds that $B_{\|\cdot\|}^{\circ}(\alpha, \vec{0}) + B_{\|\cdot\|}^{\circ}(\beta, \vec{0}) = B_{\|\cdot\|}^{\circ}(\alpha + \beta, \vec{0})$.*

The fourth fact, while also very simple, is the key change of perspective that allowed us to prove the main results of this section. It says that if we are checking which sets $X$ in our partition intersect an $\varepsilon$-ball located at $\vec{p}$ (in order to see how many there are), we can instead enlarge each member of the partition by taking its Minkowski sum with the origin-centered $\varepsilon$-ball, and check which of these enlarged members contain the point $\vec{p}$.

**Fact A.4.** *For any normed vector space, for any set $X$, for any vector $\vec{p}$, and any $\varepsilon > 0$, the following are equivalent:*

1. $B_{\|\cdot\|}^{\circ}(\varepsilon, \vec{p}) \cap X \neq \emptyset$

2. $\vec{p} \in X + B_{\|\cdot\|}^{\circ}(\varepsilon, \vec{0})$

*The same can be said replacing both open balls with closed balls.*

Now we introduce the result which is the connection to the above-mentioned key change of perspective. The result says to consider a bounded, (measurable) subset $S \subseteq \mathbb{R}^d$ (so it has finite measure) and a collection $\mathcal{A}$ of (measurable) subsets of $S$. If we compute the sum of measures of all members in the collection $\mathcal{A}$ (i.e. intuitively the total volume that they take up), and compare this to the measure/volume of $S$, then whatever this ratio is, we can find a point in $S$ covered by that many members of the collection $\mathcal{A}$. For example, in the simplest case that the total measure of members of $\mathcal{A}$ is larger than the measure of $S$, then there is no way for all of the members of $\mathcal{A}$ to be disjoint, so there has to be some point covered by two members. This simple case can be viewed as a continuous version of the pigeonhole principle.

In the more generic case, this result should be intuitively true by an averaging argument: if every point of $S$ is covered only $n$ times, then the total measure of members in $\mathcal{A}$ is at most $n \cdot m(S)$, so if the ratio of total measure in $\mathcal{A}$ to the measure of $S$ is large, then $n$ must also be large. This more general version is a sort of continuous multi-pigeonhole principle.

**Proposition A.1** (Continuous Multi-Pigeonhole Principle)**.** *Let $d \in \mathbb{N}$ and $S \subset \mathbb{R}^d$ be bounded and measurable. Let $\mathcal{A}$ be a family of measurable subsets of $S$, and let $k = \left\lceil \frac{\sum_{A \in \mathcal{A}} m(A)}{m(S)} \right\rceil$. Then if $k < \infty$, there exists $\vec{p} \in S$ such that $\vec{p}$ belongs to at least $k$ members of $\mathcal{A}$. (And if $k = \infty$, then for any $n \in \mathbb{N}$ there exists $\vec{p}^{(n)} \in S$ such that $\vec{p}^{(n)}$ belongs to at least $n$ members of $\mathcal{A}$.)*

We first encountered this result as the main ingredient in the standard proof of Blichfeldt's theorem (which was the source of motivation for our main technique), but many of the sources we found where proofs of Blichfeldt's theorem are presented did not prove the result above except in special cases, so for convenience and completion, we provide a proof in B in two parts: B.4, and B.5.

The next ingredient that we need is a way to measure how large the Minkowski sum in Fact A.4 is. In order to utilize Proposition A.1 we need a lower bound on the measures, and we can obtain one using the generalized Brunn-Minkowski inequality stated below.

**Theorem A.5** (Generalized Brunn-Minkowski Inequality)**.** *Let $d \in \mathbb{N}$ and $A, B \subseteq \mathbb{R}^d$ be Lebesgue measurable such that $A + B$ is also Lebesgue measurable. Then*

$$m(A + B) \geq \left[ m(A)^{\frac{1}{d}} + m(B)^{\frac{1}{d}} \right]^d.$$

This version of the statement can be obtained from [22, Equation 11]; in that survey, Gardner states this theorem with a requirement that the sets be bounded, but in the following paragraph notes that this is not necessary and the requirement is only stated for convenience of the presentation in that survey.

In the theorem, the requirement that $A + B$ is Lebesgue measurable is not a triviality; Gardner discusses that there exist known Lebesgue measurable sets $A$ and $B$ such that the Minkowski sum $A + B$ is not Lebesgue measurable as shown in [43]. The next result gives us a way to circumvent this issue in our application even if the members of our partition are not measurable by taking $B$ to be an open set so that the sum $A + B$ is open (and thus measurable), and using the outer measure of $A$ so that we don't need the assumption that $A$ is measurable.

**Lemma A.6.** *Let $d \in \mathbb{N}$ and let $\mathbb{R}^d$ be equipped with any norm $\|\cdot\|$. Let $Y \subseteq \mathbb{R}^d$, and $\varepsilon \in (0, \infty)$. Then $Y + B_{\|\cdot\|}^\circ(\varepsilon, \vec{0})$ is open (and thus Borel measurable), and $m(Y + B^\circ(\varepsilon, \vec{0})) \geq \left( m_{out}(Y)^{\frac{1}{d}} + \varepsilon \cdot (v_{\|\cdot\|, d})^{\frac{1}{d}} \right)^d.$*

*Proof.* By Fact A.2, for any $\varepsilon'' \in (0, \infty)$, $Y + B_{\|\cdot\|}^\circ(\varepsilon'', \vec{0}) = \bigcup_{\vec{y} \in Y} B_{\|\cdot\|}^\circ(\varepsilon'', \vec{y})$ which is a union of open sets, so is itself open and thus Borel measurable. Now, for any $\varepsilon' \in (0, \varepsilon)$, observe that by Fact A.3, $B_{\|\cdot\|}^\circ(\varepsilon, \vec{0}) = B_{\|\cdot\|}^\circ(\varepsilon - \varepsilon', \vec{0}) + B_{\|\cdot\|}^\circ(\varepsilon', \vec{0})$ and thus, this sum is measurable because it is an open ball. Using this equality and the associativity of the Minkowski sum, we have

$$Y + B_{\|\cdot\|}^\circ(\varepsilon, \vec{0}) = Y + \left[ B_{\|\cdot\|}^\circ(\varepsilon - \varepsilon', \vec{0}) + B_{\|\cdot\|}^\circ(\varepsilon', \vec{0}) \right] = \left[ Y + B_{\|\cdot\|}^\circ(\varepsilon - \varepsilon', \vec{0}) \right] + B_{\|\cdot\|}^\circ(\varepsilon', \vec{0}).$$

Thus, we have the following chain of inequalities (each justified after it is stated):

$$m\left(Y + B^\circ_{\|\cdot\|}(\varepsilon, \vec{0})\right) = m\left(\left[Y + B^\circ_{\|\cdot\|}(\varepsilon - \varepsilon', \vec{0})\right] + B^\circ_{\|\cdot\|}(\varepsilon', \vec{0})\right)$$

(Open, measurable, equality above)

$$\geq \left(m\left(Y + B^\circ_{\|\cdot\|}(\varepsilon - \varepsilon', \vec{0})\right)^{\frac{1}{d}} + m\left(B^\circ_{\|\cdot\|}(\varepsilon', \vec{0})\right)^{\frac{1}{d}}\right)^d$$

The above comes from the Theorem A.5 noting that as demonstrated above, terms of the sum $\left[Y + B^\circ_{\|\cdot\|}(\varepsilon - \varepsilon', \vec{0})\right]$ and $B^\circ_{\|\cdot\|}(\varepsilon', \vec{0})$ are open and thus measurable, and the same holds for the sum itself $\left(Y + B^\circ_{\|\cdot\|}(\varepsilon, \vec{0})\right)$. We continue.

$$\geq \left(m_{out}(Y)^{\frac{1}{d}} + m\left(B^\circ_{\|\cdot\|}(\varepsilon', \vec{0})\right)^{\frac{1}{d}}\right)^d$$

The above inequality comes from the definition of the outer measure of $Y$ being the infimum of the measures of all measurable supersets of $Y$. Since $Y \subseteq Y + B^\circ_{\|\cdot\|}(\varepsilon', \vec{0})$, we get the inequality above. Continuing, we have the following:

$$= \left(m_{out}(Y)^{\frac{1}{d}} + m\left(\varepsilon' \cdot B^\circ_{\|\cdot\|}(1, \vec{0})\right)^{\frac{1}{d}}\right)^d \quad \text{(Scaling of norm-based balls)}$$

$$= \left(m_{out}(Y)^{\frac{1}{d}} + \left[(\varepsilon')^d \cdot m\left(B^\circ_{\|\cdot\|}(1, \vec{0})\right)\right]^{\frac{1}{d}}\right)^d$$

(Scaling for Lebesgue measure)

$$= \left(m_{out}(Y)^{\frac{1}{d}} + \varepsilon' \cdot (v_{\|\cdot\|,d})^{\frac{1}{d}}\right)^d \quad \text{(Algebra and } v_{\|\cdot\|,d} \overset{\text{def}}{=} m\left(B^\circ_{\|\cdot\|}(1, \vec{0})\right))$$

Since the inequality above holds for all $\varepsilon' \in (0, \varepsilon)$, it must also hold in the limit (keeping $d$ and $Y$ fixed):

$$m\left(Y + B^\circ_{\|\cdot\|}(\varepsilon, \vec{0})\right) \geq \lim_{\varepsilon' \to \varepsilon}\left[\left(m_{out}(Y)^{\frac{1}{d}} + \varepsilon' \cdot (v_{\|\cdot\|,d})^{\frac{1}{d}}\right)^d\right] = \left(m_{out}(Y)^{\frac{1}{d}} + \varepsilon \cdot (v_{\|\cdot\|,d})^{\frac{1}{d}}\right)^d$$

which concludes the proof. $\qquad\square$

At this point, we are nearly in a position to prove the main result of this section, but we need one last result which gives an inequality that we will compose with the Theorem A.5. The result below can be interpreted as saying that for appropriate parameters, we can essentially factor our the "$x$" in $(x^{1/d} + \alpha)^d$ to get the no larger expression $x(1 + \alpha)^d$.

**Lemma A.7.** *For $d \in [1, \infty)$, $x \in [0, 1]$, and $\alpha \in [0, \infty)$, it holds that $(x^{1/d} + \alpha)^d \geq x(1 + \alpha)^d$.*

*Proof.* If $x = 0$, then the result is trivial. Otherwise $x \in (0, 1]$, so $x^{1/d} \in (0, 1]$. Because $\alpha \geq 0$, $\frac{\alpha}{x^{1/d}} \geq \alpha$, so

$$(x^{1/d} + \alpha)^d = \left(x^{1/d}\left(1 + \frac{\alpha}{x^{1/d}}\right)\right)^d = x\left(1 + \frac{\alpha}{x^{1/d}}\right)^d \geq x(1 + \alpha)^d.$$

$\qquad\square$

### A.2.2  A generic result for all Norms, and Theorem 3.1

We establish Theorem 3.1, by proving the following result.

**Theorem A.8** ($\varepsilon$-Neighborhoods for Measure Bounded Partitions and Arbitrary Norm)**.** *theorem Let $d \in \mathbb{N}$, $M \in (0, \infty)$, and $\mathcal{P}$ a partition of $\mathbb{R}^d$ such that every member has outer Lebesgue measure at most $M$. Let $\mathbb{R}^d$ be equipped with any norm $\|\cdot\|$. For every $\varepsilon \in (0, \infty)$, there exists $\vec{p} \in \mathbb{R}^d$ such that $B^\circ_{\|\cdot\|}(\varepsilon, \vec{p})$ intersects at least $k = \left\lceil \left(1 + \varepsilon \left(\frac{v_{\|\cdot\|,d}}{M}\right)^{1/d}\right)^d \right\rceil$ members of the partition where $v_{\|\cdot\|,d} \overset{\text{def}}{=} m\left(B^\circ_{\|\cdot\|}(1, \vec{0})\right)$ is the Lebesgue measure of the $\|\cdot\|$ unit ball.*

*Proof.* Throughout the proof, all lengths will be with respect to $\|\cdot\|$, so we will eliminate the clutter by neglecting to use the $\|\cdot\|$ subscript anywhere in the proof. We will also be working in a single dimension $d$ throughout the proof, so we also drop the $d$ subscript from $v$ throughout.

Consider the following for each $n \in \mathbb{N}$. Let $S_n = B^\circ(n, \vec{0})$ and $S'_n = B^\circ(n+\varepsilon, \vec{0}) = S_n + B^\circ(\varepsilon, \vec{0})$ and $\mathcal{S}$ be the partition of $S_n$ induced[2] by $\mathcal{P}$. For each $Y \in \mathcal{S}_n$, let $C_Y = Y + B^\circ(\varepsilon, \vec{0})$. By Lemma A.6, $C_Y$ is measurable, and $m(C_Y) \geq \left( m_{out}(Y)^{\frac{1}{d}} + \varepsilon \cdot v^{\frac{1}{d}} \right)^d$. Also observe that $C_Y \subseteq S'_n$. Now consider the following inequalities:

$$m(C_Y) \geq \left( m_{out}(Y)^{\frac{1}{d}} + \varepsilon \cdot v^{\frac{1}{d}} \right)^d \qquad \text{(Above)}$$

$$= \left( M^{1/d} \left[ \frac{m_{out}(Y)^{\frac{1}{d}}}{M^{1/d}} + \frac{\varepsilon \cdot v^{\frac{1}{d}}}{M^{1/d}} \right] \right)^d \qquad \text{(Algebra)}$$

$$= M \left( \left[ \frac{m_{out}(Y)}{M} \right]^{\frac{1}{d}} + \frac{\varepsilon \cdot v^{\frac{1}{d}}}{M^{1/d}} \right)^d \qquad \text{(Algebra)}$$

$$\geq M \cdot \frac{m_{out}(Y)}{M} \cdot \left( 1 + \frac{\varepsilon \cdot v^{\frac{1}{d}}}{M^{1/d}} \right)^d \qquad \text{(A.7 since } \frac{m_{out}(Y)}{M} \in [0,1])$$

$$= m_{out}(Y) \cdot \left( 1 + \frac{\varepsilon \cdot v^{\frac{1}{d}}}{M^{1/d}} \right)^d \qquad \text{(Simplify)}$$

Informally, the above shows that for each $Y \in \mathcal{S}_n$, the set $Y + B^\circ(\varepsilon, \vec{0})$ has substantially more (outer) measure than $Y$ does—specifically a factor of $\left( 1 + \frac{\varepsilon \cdot v^{\frac{1}{d}}}{M^{1/d}} \right)^d$. We will extend this to unsurprisingly show that this implies that $\left\{ Y + B^\circ(\varepsilon, \vec{0}) \right\}_{Y \in \mathcal{S}_n}$ also has this same factor more (outer) measure than $\mathcal{S}_n$ does, observing that $\mathcal{S}_n$ has total (outer) measure of about $m(S_n)$ since $\mathcal{S}_n$ is a partition of $S_n$ (any discrepancy is due to non-measurable members in $\mathcal{S}_n$)

Formally, we claim that there exists a finite subfamily $\mathcal{F}_n \subseteq \mathcal{S}_n$ such that

$$\sum_{Y \in \mathcal{F}_n} m\left( Y + B^\circ(\varepsilon, \vec{0}) \right) \geq \left( 1 + \frac{\varepsilon \cdot v^{\frac{1}{d}}}{M^{1/d}} \right)^d \cdot m(S_n).$$

To see this, first consider the case that $\mathcal{S}_n$ has infinite cardinality. Let $\mathcal{F}_n \subset \mathcal{S}_n$ be any subfamily of finite cardinality at least $\left( 1 + \frac{\varepsilon \cdot v^{\frac{1}{d}}}{M^{1/d}} \right)^d \cdot m(S_n) \cdot \frac{1}{\varepsilon^d v}$. This gives

$$\sum_{Y \in \mathcal{F}_n} m\left( Y + B^\circ(\varepsilon, \vec{0}) \right) \geq \sum_{Y \in \mathcal{F}_n} m\left( B^\circ(\varepsilon, \vec{0}) \right)$$

---

[2]I.e. $\mathcal{S} = \{X \cap S_n : X \in \mathcal{P} \text{ and } X \cap S_n \neq \emptyset\}$. That is, $\mathcal{S}$ is the partition of $S_n$ obtained by intersecting every member of $\mathcal{P}$ with the new domain $S_n$ and keeping those that have non-empty intersections.

where this inequality is because $Y + B^\circ(\varepsilon, \vec{0})$ is a superset of some translation of $B^\circ(\varepsilon, \vec{0})$ since $Y \neq \emptyset$. Continuing, we use the standard fact that $m\left(B^\circ(\varepsilon, \vec{0})\right) = m\left(\varepsilon \cdot B^\circ(1, \vec{0})\right) = \varepsilon^d \cdot m\left(B^\circ(1, \vec{0})\right) = \varepsilon^d v$:

$$\geq \sum_{Y \in \mathcal{F}_n} \varepsilon^d v$$

$$= \left[\left(1 + \frac{\varepsilon \cdot v^{\frac{1}{d}}}{M^{1/d}}\right)^d \cdot m(S_n) \cdot \frac{1}{\varepsilon^d v}\right] \cdot \varepsilon^d v \qquad \text{(Cardinality of } \mathcal{F}_n)$$

$$= \left(1 + \frac{\varepsilon \cdot v^{\frac{1}{d}}}{M^{1/d}}\right)^d \cdot m(S_n). \qquad \text{(Simplify)}$$

Now consider the other (more interesting) case where $\mathcal{S}_n$ has finite cardinality[3]. Take $\mathcal{F}_n = \mathcal{S}_n$ so that

$$\sum_{Y \in \mathcal{F}_n} m\left(Y + B^\circ(\varepsilon, \vec{0})\right) = \sum_{Y \in \mathcal{S}_n} m\left(Y + B^\circ(\varepsilon, \vec{0})\right) \qquad (\mathcal{F}_n = \mathcal{S}_n)$$

$$\geq \sum_{Y \in \mathcal{S}_n} m_{out}(Y) \cdot \left(1 + \frac{\varepsilon \cdot v^{\frac{1}{d}}}{M^{1/d}}\right)^d \qquad \text{(Above)}$$

$$= \left(1 + \frac{\varepsilon \cdot v^{\frac{1}{d}}}{M^{1/d}}\right)^d \cdot \sum_{Y \in \mathcal{S}_n} m_{out}(Y) \qquad \text{(Linearity of summation)}$$

$$\geq \left(1 + \frac{\varepsilon \cdot v^{\frac{1}{d}}}{M^{1/d}}\right)^d \cdot m_{out}\left(\bigcup_{Y \in \mathcal{S}_n} Y\right)$$

where the above inequality is due the the countable subaddativity property of outer measures which states that a countable sum of outer measures of sets is at least as large as the outer measure of the union of the sets. In the last step we get

$$= \left(1 + \frac{\varepsilon \cdot v^{\frac{1}{d}}}{M^{1/d}}\right)^d \cdot m(S_n) \qquad (\bigsqcup_{Y \in \mathcal{F}_n} Y = S_n \text{ is measurable})$$

Thus, regardless of whether $\mathcal{S}_n$ has infinite or finite cardinality, there exists a finite subfamily $\mathcal{F}_n \subseteq \mathcal{S}_n$ such that

$$\sum_{Y \in \mathcal{F}_n} m\left(Y + B^\circ(\varepsilon, \vec{0})\right) \geq \left(1 + \frac{\varepsilon \cdot v^{\frac{1}{d}}}{M^{1/d}}\right)^d \cdot m(S_n).$$

Fix such a subfamily $\mathcal{F}_n$, and let $\mathcal{A}_n = \left\{Y + B^\circ(\varepsilon, \vec{0})\right\}_{Y \in \mathcal{F}_n}$ be a family indexed[4] by $\mathcal{F}_n$. Note that for each $A_Y \overset{\text{def}}{=} Y + B^\circ(\varepsilon, \vec{0}) \in \mathcal{A}_n$, since $Y \subseteq S_n = B^\circ(n, \vec{0})$, we have $A_Y \subseteq S_n + B^\circ(\varepsilon, \vec{0}) = S_n'$.

---

[3]In fact this case also works if $\mathcal{S}_n$ is countable even though we have already dealt with that case.

[4]We require this to be an indexed family rather than just a set, because it is possible that there are distinct $Y, Y' \in \mathcal{S}_n$ such that $Y + B^\circ(\varepsilon, \vec{0}) = Y' + B^\circ(\varepsilon, \vec{0})$.

Thus, by B.5[5], there is a point $\vec{p}^{(n)} \in S'_n$ which belongs to at least $k_n$-many sets in $\mathcal{A}_n$ where

$$k_n \overset{\text{def}}{=} \left\lceil \frac{\sum_{Y \in \mathcal{F}_n} m\left(Y + B^\circ(\varepsilon, \vec{0})\right)}{m(S'_n)} \right\rceil \geq \frac{\left(1 + \frac{\varepsilon \cdot v^{\frac{1}{d}}}{M^{1/d}}\right)^d \cdot m(S_n)}{m(S'_n)} \quad \text{(Above)}$$

$$= \left(1 + \frac{\varepsilon \cdot v^{\frac{1}{d}}}{M^{1/d}}\right)^d \cdot \frac{m\left(B^\circ(n, \vec{0})\right)}{m\left(B^\circ(n+\varepsilon, \vec{0})\right)}$$

$$\text{(Def'n of } S_n \text{ and } S'_n)$$

$$= \left(1 + \frac{\varepsilon \cdot v^{\frac{1}{d}}}{M^{1/d}}\right)^d \cdot \frac{n^d \cdot v}{(n+\varepsilon)^d \cdot v} \quad \text{(Standard scaling fact)}$$

$$= \left(1 + \frac{\varepsilon \cdot v^{\frac{1}{d}}}{M^{1/d}}\right)^d \cdot \left(\frac{n}{n+\varepsilon}\right)^d. \quad \text{(Simplify)}$$

Since $\vec{p}^{(n)}$ belongs to at least $k_n$-many sets in $\mathcal{A}_n$, this by definition means that there are at least $k_n$-many sets $Y \in \mathcal{F}_n$ such that $\vec{p}^{(n)} \in Y + B^\circ(\varepsilon, \vec{0})$, so by Fact A.4, we have $Y \cap B^\circ(\varepsilon, \vec{p}^{(n)}) \neq \emptyset$. For each such $Y$ (since $Y \in \mathcal{F}_n \subseteq \mathcal{S}_n$), there is a distinct[6] $X_Y \in \mathcal{P}$ such that $Y \subseteq X_Y$ and thus $X_Y \cap B^\circ(\varepsilon, \vec{0}) \neq \emptyset$ showing that there are at least $k_n$-many sets in $\mathcal{P}$ which intersect $B^\circ(\varepsilon, \vec{0})$.

For the last step of the proof, we perform a limiting process on $n$. By Fact A.1, let $\gamma \in \mathbb{R}$ such that $\gamma < \left(1 + \frac{\varepsilon \cdot v^{\frac{1}{d}}}{M^{1/d}}\right)^d$ and $\lceil \gamma \rceil = \left\lceil \left(1 + \frac{\varepsilon \cdot v^{\frac{1}{d}}}{M^{1/d}}\right)^d \right\rceil$. Then, because

$$\lim_{n \to \infty} k_n \geq \lim_{n \to \infty} \left(1 + \frac{\varepsilon \cdot v^{\frac{1}{d}}}{M^{1/d}}\right)^d \cdot \left(\frac{n}{n+\varepsilon}\right)^d = \left(1 + \frac{\varepsilon \cdot v^{\frac{1}{d}}}{M^{1/d}}\right)^d > \gamma,$$

we can take $N \in \mathbb{N}$ sufficiently large so that

$$k_N \geq \left(1 + \frac{\varepsilon \cdot v^{\frac{1}{d}}}{M^{1/d}}\right)^d \cdot \left(\frac{N}{N+\varepsilon}\right)^d > \gamma.$$

Then considering the point $\vec{p}^{(N)}$, we have by the choice of $\gamma$ and the fact that $k_N$ is an integer that

$$k_N = \lceil k_N \rceil \geq \lceil \gamma \rceil = \left\lceil \left(1 + \frac{\varepsilon \cdot v^{\frac{1}{d}}}{M^{1/d}}\right)^d \right\rceil$$

showing that $B^\circ(\varepsilon, \vec{p}^{(N)})$ intersects at least $k_N \geq \left\lceil \left(1 + \frac{\varepsilon \cdot v^{\frac{1}{d}}}{M^{1/d}}\right)^d \right\rceil$ members of $\mathcal{P}$ as claimed which completes the proof. $\square$

Now Theorem 3.1 follows as a simple corollary of A.8.

*Proof.* Consider the $\ell_\infty$ norm and $M = 1$ noting that for each $d \in \mathbb{N}$, $v_{\|\cdot\|_\infty, d} = 2^d$ (i.e. the volume of the $\ell_\infty$ unit ball in $\mathbb{R}^d$ is $2^d$).

Then by A.8, there is a point $\vec{p} \in \mathbb{R}^d$ such that $B^\circ_\infty(\varepsilon, \vec{p})$ intersects at least

$$\left(1 + \varepsilon \left(\frac{v_{\|\cdot\|_\infty, d}}{M}\right)^{1/d}\right)^d = \left(1 + \varepsilon \left(\frac{2^d}{1}\right)^{1/d}\right)^d = (1 + 2\varepsilon)^d$$

---

[5]We are taking $X$ in B.5 to be $S'_n$ in this proof, and taking $\mu$ to be $m$ and taking $\mathcal{A}$ to be $\mathcal{A}_n$. We have that $\sum_{A \in \mathcal{A}_n} m(A) < \infty$ because $|\mathcal{A}_n| = |\mathcal{F}_n|$ is finite, and each $A \in \mathcal{A}_n$ is a subset of $S'_n$, so has finite measure.

[6]I.e. for $Y \neq Y' \in \mathcal{S}_n$ we have that $X_Y, X_{Y'} \in \mathcal{P}$ and $X_Y \neq X_{Y'}$ so this mapping of $Y$'s to $X$'s is injective, so we have at least the same cardinality of $X$'s with the desired property as $Y$'s with the desired property.

members of $\mathcal{P}$, and thus trivially the closed ball $\overline{B}_\infty(\varepsilon, \vec{p})$ does as well. Thus, if $\mathcal{P}$ is $(k, \varepsilon)$-secluded (meaning by definition that for every $\vec{p} \in \mathbb{R}^d$ it holds that $\overline{B}_\infty(\varepsilon, \vec{p})$ intersects at most $k$ members of $\mathcal{P}$) then it must be that $k \geq (1 + 2\varepsilon)^d$.

For the consequence, if $k \leq 2^d$, then this implies $\varepsilon \leq \frac{1}{2}$. Then taking the logarithm of both sides of our inequality and using the fact that $\log_4(1 + 2x) \geq x$ for $x \in [0, \frac{1}{2}]$ (see footnote[7]), we have

$$\log_4(k) \geq d \log_4(1 + 2\varepsilon) \geq d\varepsilon$$

showing that $\varepsilon \leq \frac{\log_4(k)}{d}$. $\qquad\square$

We state the following interesting corollary when $k(d)$ is polynomial.

**Corollary A.9.** *Let $\langle \mathcal{P}_d \rangle_{d=1}^\infty$ be a sequence of $(k(d), \varepsilon(d))$-secluded partitions of $\mathbb{R}^d$ such that every member of each $\mathcal{P}_d$ has outer Lebesgue measure at most 1. If $k(d) \in \mathsf{poly}(d)$ then $\varepsilon(d) \in O(\frac{\ln d}{d})$ (where the hidden constant can be taken to be anything exceeding the polynomial degree of $k$).*

*Proof.* Since $k(d) \in \mathsf{poly}(d)$, then there are constants $C, n$ such that for sufficiently large $d$, we have $k(d) \leq Cd^n$ which for sufficiently large $d$ is less than $2^d$ so by 3.1, for sufficiently large $d$ we have

$$\varepsilon(d) \leq \frac{\ln(k(d))}{d} \leq \frac{n \ln(Cd)}{d} \in O\left(\frac{\ln(d)}{d}\right).$$

More specifically, for any $n' > n$ we have for large enough $d$ that $(n' - n)\ln(d) \geq n \ln(C)$, so for large enough $d$ we have

$$\varepsilon(d) \leq \frac{n \ln(Cd)}{d} = \frac{n[\ln(C) + \ln(d)]}{d} \leq \frac{(n' - n)\ln(d) + n \ln(d)}{d} = \frac{n' \ln(d)}{d}$$

showing that the hidden constant can be taken to be any $n'$ larger than the degree $n$ of $k$. $\qquad\square$

## A.3 A Family of Near Optimal Constructions

The partitions that we construct in this section are of a very natural form: we build a partition of a large dimension $d$ space, by splitting up the coordinates into smaller sets, and separately partitioning each set of coordinates. In the end, the smaller partitions will be known partition constructions [44]. We will define the construction very generically. We need two basic results. The following observation notes that if a partition is $(k, \varepsilon)$-secluded, then we can *increase* $k$ to $k'$ and *decrease* $\varepsilon$ to $\varepsilon'$ and the partition is trivially $(k', \varepsilon')$-secluded.

**Observation A.10** (Monotonicity in $k$ and $\varepsilon$). *Let $d \in \mathbb{N}$, $k, k' \in \mathbb{N}$ with $k' \geq k$, $\varepsilon, \varepsilon' \in [0, \infty)$ with $\varepsilon' \leq \varepsilon$, and $\mathcal{P}$ a $(k, \varepsilon)$-secluded partition of $\mathbb{R}^d$. Then $\mathcal{P}$ is also a $(k', \varepsilon')$-secluded partition of $\mathbb{R}^d$.*

*Proof.* Since $\mathcal{P}$ is $(k, \varepsilon)$-secluded, by definition every $\varepsilon$-ball intersects at most $k$ members of $\mathcal{P}$, so trivially every (no larger) $\varepsilon'$-ball intersects at most $k' \geq k$ members of $\mathcal{P}$. $\qquad\square$

We will frequently refer to the above observation just using the phrase "by monotonicity, $\mathcal{P}$ is $(k', \varepsilon')$-secluded"

**Fact A.11** (Trivial $k$ for Unit Cube Partitions). *Let $d \in \mathbb{N}$, $\varepsilon \in [0, \infty)$, and $\mathcal{P}$ be any unit cube partition of $\mathbb{R}^d$. Then $\mathcal{P}$ is $(k, \varepsilon)$-secluded for $k = \lfloor (2 + 2\varepsilon)^d \rfloor$.*

*Proof.* Consider any point $\vec{p} \in \mathbb{R}^d$. Observe that for any $X \in \mathcal{P}$, $X$ is a unit cube, so $\mathrm{diam}_\infty(X) = 1$, so if $X$ intersects $\overline{B}_\infty(\varepsilon, \vec{p})$, then $X \subseteq \overline{B}_\infty(1 + \varepsilon, \vec{p})$.

Because (1) each $X \in \mathcal{P}$ has measure 1, and (2) every pair of members are disjoint (because $\mathcal{P}$ is a partition), and (3) the measure of $\overline{B}_\infty(1 + \varepsilon, \vec{p}) = \vec{p} + [-1 - \varepsilon, 1 + \varepsilon]^d$ is $(2 + 2\varepsilon)^d$, it follows that at most $\lfloor (2 + 2\varepsilon)^d \rfloor$ members of $\mathcal{P}$ are a subset of $\overline{B}_\infty(1 + \varepsilon, \vec{p})$ and thus at most $\lfloor (2 + 2\varepsilon)^d \rfloor$ members of $\mathcal{P}$ intersect $\overline{B}_\infty(\varepsilon, \vec{p})$ which shows that $\mathcal{P}$ is $(k, \varepsilon)$-secluded for $k = \lfloor (2 + 2\varepsilon)^d \rfloor$ as claimed. $\qquad\square$

---

[7]One can verify that the function $\log_4(1 + 2x) - x$ is concave on its domain $(-\frac{1}{2}, \infty)$ using the second derivative and note that it is 0 at $x = 0$ and $x = \frac{1}{2}$ (which is why the base 4 logarithm was chosen) so it is non-negative on $[0, \frac{1}{2}]$.

### A.3.1 Construction

**Definition A.12** (Partition Product). *Let $d_1, \ldots, d_n \in \mathbb{N}$ and $\mathcal{P}_1, \ldots, \mathcal{P}_n$ be partitions of $\mathbb{R}^{d_1}, \ldots, \mathbb{R}^{d_n}$ respectively. Letting $d = \sum_{i=1}^{n} d_n$ we define the product partition of $\mathbb{R}^d$ as*

$$\prod_{i=1}^{n} \mathcal{P}_i \stackrel{\text{def}}{=} \left\{ \prod_{i=1}^{n} X_i : X_i \in \mathcal{P}_i \right\}$$

*where $\prod_{i=1}^{n} X_i$ is viewed as a subset of $\mathbb{R}^d$.*

We specifically stated that $\prod_{i=1}^{n} X_i$ is viewed as a subset of $\mathbb{R}^d$, because technically it is a subset of $\prod_{i=1}^{n} \mathbb{R}^{d_i}$, but this is naturally isomorphic to $\mathbb{R}^d = \mathbb{R}^{\sum_{i=1}^{n} d_i}$. For example, technically, if $d_1 = d_2 = d_3 = 2$, then the elements of $\prod_{i=1}^{n} \mathbb{R}^{d_i}$ are of the form $\langle \langle x_1, x_2 \rangle, \langle x_3, x_4 \rangle, \langle x_5, x_6 \rangle \rangle$, but this is trivially isomorphic to $\mathbb{R}^6$ by instead considering the element as $\langle x_1, x_2, x_3, x_4, x_5, x_6 \rangle$.

Also observe (shown below) that if the original partitions were unit cube partitions, then the product partition is also a unit cube partition.

**Fact A.13** (Unit Cube Preservation). *If $d_1, \ldots, d_n \in \mathbb{N}$ and $\mathcal{P}_1, \ldots, \mathcal{P}_n$ are unit cube partitions of $\mathbb{R}^{d_1}, \ldots, \mathbb{R}^{d_n}$ respectively, then $\prod_{i=1}^{n} \mathcal{P}_i$ is also a unit cube partition.*

*Proof.* Each member of $\prod_{i=1}^{n} \mathcal{P}_i$ is of the form $\prod_{i=1}^{n} X_i$ where $X_i \in \mathcal{P}_i$. Since $\mathcal{P}_i$ is a unit cube partition, each $X_i$ is a product of translates of $[0, 1)$, and thus $\prod_{i=1}^{n} X_i$ is also a product of translates of $[0, 1)$, so the member is a unit cube. $\qquad\square$

If all partitions $\mathcal{P}_1, \ldots, \mathcal{P}_n$ are "efficiently computable" in the sense that given an arbitrary point, $\vec{x} \in \mathbb{R}^{d_i}$ there is an efficient algorithm that computes a representation of the member of $\mathcal{P}_i$ containing $\vec{x}$, then the product partition is also "efficiently computable" because given some point $\vec{y} \in \mathbb{R}^d$, the member that it is contained in can be found by determining which member of $\mathcal{P}_1$ the point $\langle y_i \rangle_{i=1}^{d_1}$ is in, and independently determining which member of $\mathcal{P}_2$ the point $\langle y_i \rangle_{i=d_1+1}^{d_1+d_2}$ is in, etc. The member of $\prod_{i=1}^{d} \mathcal{P}_i$ that contains $\vec{y}$ is just the product of members. This is an important property for using partitions as the basis of rounding schemes because an algorithm must determine which member/equivalence class a point is in (even if just implicitly). Because the partitions of [44] are "efficiently computable" so are the partitions constructed here.

We can now show that if we take a product of partitions, and we have a guarantee for each $\mathcal{P}_i$ that it is $(k_i, \varepsilon_i)$-secluded, then we can guarantee the product partition is $(k, \varepsilon)$-secluded where $k$ is the product of the $k_i$'s and $\varepsilon$ is the minimum of the $\varepsilon_i$'s.

**Proposition A.2** (Product Partition Seclusion Guarantees). *Let $n \in \mathbb{N}$. For each index $i \in [n]$, let $d_i, k_i \in \mathbb{N}$, $\varepsilon_i \in (0, \infty)$ and $\mathcal{P}_i$ be a $(k_i, \varepsilon_i)$-secluded partition of $\mathbb{R}^{d_i}$. Then the product partition $\mathcal{P} = \prod_{i=1}^{n} \mathcal{P}_i$ is a $(k, \varepsilon)$-secluded partition of $\mathbb{R}^d$ where $d = \sum_{i=1}^{n} d_i$, and $k = \prod_{i=1}^{n} k_i$, and $\varepsilon = \min_{i \in [n]} \varepsilon_i$.*

*Proof Sketch.* The basic idea is that for any point $\vec{p} \in \mathbb{R}^d$, we consider how many members of $\mathcal{P}$ intersect $\overline{B}_\varepsilon(\vec{p})$. Conceptually[8], we think of $\vec{p}$ as a sequence $\langle \vec{p}^{(i)} \rangle_{i=1}^{n}$ of $n$ points where the $i$th point $\vec{p}^{(i)}$ belongs to $\mathbb{R}^{d_i}$. Because we are working with the $\ell_\infty$ norm (that is the norm used by the definition of secluded), the $\varepsilon$ ball around $\vec{p}$ is the product of the $\varepsilon$ balls around each $\vec{p}^{(i)}$ which is smaller than the product of $\varepsilon_i$ balls around each $\vec{p}^{(i)}$ because we chose $\varepsilon$ as the minimum size. Thus, if the $\varepsilon$ ball around $\vec{p}$ intersects a member $X$ of the partition $\mathcal{P}$, then conceptually viewing $X$ as a sequence $\langle X_i \rangle_{i=1}^{n}$ where $X_i$ is a member of $\mathcal{P}_i$, it must be for each $i \in [n]$ that the $\varepsilon$ ball around $\vec{p}^{(i)}$ intersects $X_i$ (and thus so does the $\varepsilon_i$ ball since $\varepsilon_i \geq \varepsilon$). This means (for each $i \in [n]$) that $X_i$ is one of at most $k_i$ members of $\mathcal{P}_i$ because at most $k_i$ members of $\mathcal{P}_i$ intersect the $\varepsilon_i$ ball around $\vec{p}^{(i)}$ (by definition of $\mathcal{P}_i$ being $(k_i, \varepsilon_i)$-secluded). Thus $X$ is one of at most $\prod_{i=1}^{n} k_i = k$ members of $\mathcal{P}$. That is, there are at most $k$ members of $\mathcal{P}$ that intersect the $\varepsilon$ ball around $\vec{p}$ which is the definition of $\mathcal{P}$ being $(k, \varepsilon)$-secluded. $\qquad\square$

---

[8]In other words we identify the set $\mathbb{R}^d$ with $\mathbb{R}^{d_1} \times \mathbb{R}^{d_2} \times \cdots \times \mathbb{R}^{d_{n-1}} \times \mathbb{R}^{d_n}$

Utilizing the construction above, we will now take a unit cube partition of [44] for each $\mathbb{R}^{d_i}$ and take the product to obtain a new partition. Since the dimension of each $d_i$ is smaller than the dimension $d$, this allows us to get a larger value of $\varepsilon_i$ for each partition, and thus a larger value of $\varepsilon$ for the partition of $\mathbb{R}^d$ than if we had used one of the original partitions. The price we pay for this is that the value of $k$ also increases.

We establish the following theorem stated as Theorem 5.3 in the main body of the paper.

**Theorem A.14.** *Let $d \in \mathbb{N}$ and $d' \in [d]$. There exists a $(k, \varepsilon)$-secluded unit cube partition of $\mathbb{R}^d$ where $k = (d'+1)^{\lceil \frac{d}{d'} \rceil}$ and $\varepsilon = \frac{1}{2d'}$.*

*Proof.* Let $n = \lceil \frac{d}{d'} \rceil$. By [44], let $\mathcal{P}'$ be a $(d'+1, \frac{1}{2d'})$-secluded unit cube partition of $\mathbb{R}^{d'}$.

By Proposition A.2 and Fact A.13, $\mathcal{P} = \prod_{i=1}^{n} \mathcal{P}'$ is a $(k, \varepsilon)$-secluded unit cube partition of $\mathbb{R}^{n \cdot d'}$ where $k = (d'+1)^n$ and $\varepsilon = \frac{1}{2d'}$. Since $n \cdot d' = \lceil \frac{d}{d'} \rceil \cdot d' \geq d$, this trivially (by ignoring extra coordinates) gives a partition of $\mathbb{R}^d$ with these same properties (alternatively, see footnote[9]) which completes the proof. $\square$

This construction is sufficiently general that for any dimension $d \in \mathbb{N}$ and any tolerance parameter in the domain of interest ($k \in [2^d]$), we can construct a $(k, \varepsilon)$-secluded unit cube partition of $\mathbb{R}^d$ such that $\varepsilon$ which is within a factor of $8 \log_4(d+1)$ of the maximum possible tolerance for this choice of dimension and degree.

Finally, we are ready to prove Theorem 3.2.

*Proof.* Let $d' \in [d]$ be the minimum integer such that $(d'+1)^{\lceil \frac{d}{d'} \rceil} \leq k$ (see justification[10]), and let $\varepsilon = \frac{1}{2d'}$. By Theorem A.14 and monotonicity, let $\mathcal{P}$ be a $(k, \varepsilon)$-secluded unit cube partition of $\mathbb{R}^d$. Now we prove the bound on $\varepsilon$ in two cases: either $d' = 1$ or $d' \geq 2$.

In the case that $d' = 1$, then $\varepsilon = \frac{1}{2}$ and $k \geq (d'+1)^{\lceil \frac{d}{d'} \rceil} = 2^d$ and by hypothesis $k \leq 2^d$, so we have equality and we conclude that

$$\frac{\log_4(k)}{4d \log_4(d+1)} = \frac{d/2}{4d \log_4(d+1)} = \frac{1}{8 \log_4(d+1)} \leq \frac{1}{8 \cdot \frac{1}{2}} \leq \frac{1}{2} = \varepsilon$$

which proves the bound on $\varepsilon$ in the first case.

In the other case, we have $d \geq d' \geq 2$. Let $d'' = d' - 1 > 0$ and $\delta = \frac{1}{2d''}$. Note that

$$\frac{\varepsilon}{\delta} = \frac{2d''}{2d'} = \frac{d'-1}{d'} = 1 - \frac{1}{d'} \geq 1 - \frac{1}{2} = \frac{1}{2}$$

so $\varepsilon \geq \frac{1}{2}\delta$. By our choice of $d'$ and because $d'' < d'$, it must be that $k \leq (d''+1)^{\lceil \frac{d}{d''} \rceil}$. Noting that $\lceil \frac{d}{d''} \rceil \leq \frac{d}{d''} + 1 = \frac{d+d''}{d''} \leq \frac{2d}{d''}$, we have

$$k \leq (d''+1)^{\lceil \frac{d}{d''} \rceil} \leq (d''+1)^{\frac{2d}{d''}} = (d''+1)^{4d\delta} \leq (d''+1)^{8d\varepsilon} \leq (d+1)^{8d\varepsilon}$$

By taking the logarithm[11], of each side and solving for $\varepsilon$, we obtain the desired result:

$$\varepsilon \geq \frac{\log_4(k)}{8d \log_4(d+1)}.$$

$\square$

---

[9] An alternate perspective is to let $d_1, \ldots, d_n$ be such that $\sum_{i=1}^{n} d_i = d$ and the first portion of the list $d_i = d'$, and the second portion of the list $d_i = d'' \overset{\text{def}}{=} d' - 1$. Then let $\mathcal{P}'$ a $(d'+1, \frac{1}{2d'})$-secluded partition of $\mathbb{R}^{d'}$ as before, and let $\mathcal{P}''$ a $(d''+1, \frac{1}{2d''})$-secluded partition of $\mathbb{R}^{d''}$. Since $d'' < d'$, $\mathcal{P}''$ is (by monotonicity) a $(d'+1, \frac{1}{2d'})$-secluded partition. Then take $\mathcal{P}_i = \mathcal{P}'$ when $d_i = d'$ and $\mathcal{P}_i = \mathcal{P}''$ when $d_i = d''$. Again, we get that $\mathcal{P}$ is $(k, \varepsilon)$-secluded for $k = (d'+1)^n$ and $\varepsilon = \frac{1}{2d'}$.

[10] A minimum exists by finiteness and the fact that some $d'$ satisfies the condition—in particular $d' = d$ satisfies the requirement because $k \geq d + 1$ by hypothesis.

[11] We choose the base 4 logarithm since that is what appears in our upper bound on $\varepsilon$.

## A.4 A Neighborhood Sperner/KKM Lemma: Theorem 3.4

In this section, we restate and prove our neighborhood variant of the Sperner/KKM/Lebesgue result on the cube. The proof idea is illustrated in Figure 2.

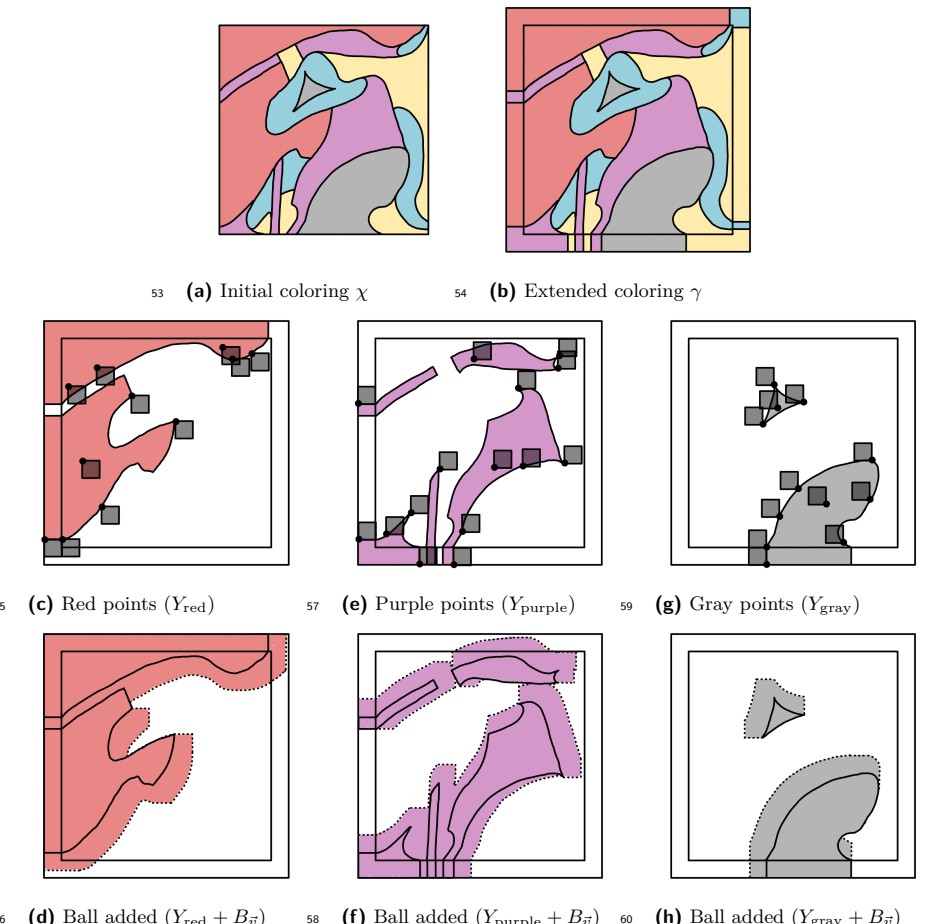

53 **(a)** Initial coloring $\chi$      54 **(b)** Extended coloring $\gamma$

55 **(c)** Red points ($Y_{\text{red}}$)    57 **(e)** Purple points ($Y_{\text{purple}}$)    59 **(g)** Gray points ($Y_{\text{gray}}$)

56 **(d)** Ball added ($Y_{\text{red}} + B_{\vec{v}}$)    58 **(f)** Ball added ($Y_{\text{purple}} + B_{\vec{v}}$)    60 **(h)** Ball added ($Y_{\text{gray}} + B_{\vec{v}}$)

Figure 2: (a) shows a (finite) coloring $\chi$ of the unit cube $[-\frac{1}{2}, \frac{1}{2}]^2$ such that no color includes points on opposite edges. (b) shows the natural extension $\gamma$ of that coloring to $[-\frac{1}{2} - \varepsilon, \frac{1}{2} + \varepsilon]^2$. The extension is obtained by mapping each point $\vec{y}$ to the point $\vec{x}$ for which each coordinate value $y_i$ is restricted to be within $[-\frac{1}{2}, \frac{1}{2}]$, and then $\vec{y}$ is given whatever color $\vec{x}$ had. (c), (e), and (g) show three of the five colors and demonstrate that there is at least one quadrant of the $\varepsilon$-ball that can be Minkowski summed with the color so that the sum remains a subset of the extended cube. For red it is the lower right quadrant, for purple it is the upper right, and for gray it could be the upper left (shown) or the upper right. (d), (f), and (h) show the resulting Minkowski sum for each color. Utilizing the Brunn-Minkowski inequality, this set will have substantially greater area—by a factor of at least $\left(1 + \frac{\varepsilon}{1+\varepsilon}\right)^2$.

**Theorem A.15** (Neighborhood Sperner/KKM Lemma). *Given a Sperner/KKM coloring of $[0, 1]^d$, for any $\varepsilon \in (0, \frac{1}{2}]$, there exists a point $\vec{p} \in [0, 1]^d$ such that $B_\infty^\circ(\varepsilon, \vec{p})$ contains at least $\left(1 + \frac{2}{3}\varepsilon\right)^d$ points with distinct colors.*

*Proof.* For convenience, we will assume that the cube is $[-\frac{1}{2}, \frac{1}{2}]^d$ rather than $[0, 1]^d$. Let $C$ be a finite set (of colors) and $\chi \colon [-\frac{1}{2}, \frac{1}{2}]^d \to C$ be a Sperner/KKM coloring of the unit cube $[-\frac{1}{2}, \frac{1}{2}]^d$.

The first step in the proof is to extend the coloring $\chi$ to the larger cube $[-\frac{1}{2} - \varepsilon, \frac{1}{2} + \varepsilon]^d$ in a natural way. Consider the following function $f$ which truncates points in the larger interval to be in the smaller interval:

$$f \colon [-\tfrac{1}{2} - \varepsilon, \tfrac{1}{2} + \varepsilon] \to [-\tfrac{1}{2}, \tfrac{1}{2}]$$

$$f(y) \stackrel{\text{def}}{=} \begin{cases} -\frac{1}{2} & y \leq -\frac{1}{2} \\ y & y \in (-\frac{1}{2}, \frac{1}{2}) \\ \frac{1}{2} & y \geq \frac{1}{2} \end{cases}$$

Let $\vec{f} \colon [-\frac{1}{2} - \varepsilon, \frac{1}{2} + \varepsilon]^d \to [-\frac{1}{2}, \frac{1}{2}]^d$ be the function which is $f$ in each coordinate: $\vec{f}(\vec{y}) \stackrel{\text{def}}{=} \langle f(y_i) \rangle_{i=1}^d$.

Now extend the coloring $\chi$ to the coloring $\gamma \colon [-\frac{1}{2} - \varepsilon, \frac{1}{2} + \varepsilon]^d \to C$ defined by

$$\gamma(\vec{x}) \stackrel{\text{def}}{=} \chi\left(\vec{f}(\vec{x})\right).$$

For each color $c \in C$, let $Y_c = \gamma^{-1}(c)$ denote the points assigned color $c$ by $\gamma$ and note that $Y_c \supseteq X_c$. Consistent with this notation, we will typically refer to a point in the unit cube as $\vec{x}$ and a point in the extended cube as $\vec{y}$.

We make the following claim which implies that for each color $c \in C$, the set $Y_c$ of points of that color in the extended coloring are contained in a set bounded away from one side of the extended cube $[-\frac{1}{2} - \varepsilon, \frac{1}{2} + \varepsilon]^d$ in each coordinate.

**Subclaim A.16.** *For each color $c \in C$ there exists an orientation $\vec{v} \in \{-1, 1\}^d$ such that $Y_c \subseteq \prod_{i=1}^d v_i \cdot (-\frac{1}{2}, \frac{1}{2} + \varepsilon]$.*

*Proof of Claim.* Fix an arbitrary coordinate $i \in [d]$. Note that for every $\vec{y} \in Y_c$ we have $f(\vec{y} \in X_c$ which is to say that the $\vec{y}$ has the same color in the extended coloring as $f(\vec{y})$ does in the original coloring (see justification[12]).

Note that if there is some $\vec{y} \in Y_c$ with $y_i \leq -\frac{1}{2}$, then $f(y_i) = -\frac{1}{2}$ so $\pi_i(X_c) \ni f(y_i) = -\frac{1}{2}$. Similarly, if there is some $\vec{y} \in Y_c$ with $y_i \geq \frac{1}{2}$, then $\pi_i(X_c) \ni \frac{1}{2}$. Recall that by hypothesis, $\pi_i(X_c)$ does not contain both $-\frac{1}{2}$ and $\frac{1}{2}$ which means it is either the case that for all $\vec{y} \in Y_c$ we have $y_i > -\frac{1}{2}$ (so $\pi_i(Y_c) \subseteq (-\frac{1}{2}, \frac{1}{2} + \varepsilon]$) or it is the case that for all $\vec{y} \in Y_c$ we have $y_i < \frac{1}{2}$ (so $\pi_i(Y_c) \subseteq [-\frac{1}{2} - \varepsilon, \frac{1}{2})$).

Thus we can choose $v_i \in \{-1, 1\}$ such that $\pi_i(Y_c) \subseteq v_i \cdot (-\frac{1}{2}, \frac{1}{2} + \varepsilon]$. Since this is true for each coordinate $i \in [d]$ we can select $\vec{v} \in \{-1, 1\}^d$ such that

$$Y_c \subseteq \prod_{i=1}^d \pi_i(Y_c) \subseteq \prod_{i=1}^d v_i \cdot (-\frac{1}{2}, \frac{1}{2} + \varepsilon]$$

as claimed. ∎

For an orientation $\vec{v} \in \{-1, 1\}^d$, let $B_{\vec{c}}$ denote the set $B_{\vec{v}} \stackrel{\text{def}}{=} \prod_{i=1}^d -v_i \cdot (0, \varepsilon)$ which should be interpreted as an open orthant of the $\ell_\infty$ $\varepsilon$-ball centered at the origin—specifically the orthant opposite the orientation $\vec{v}$. Building on A.16, we get the following:

**Subclaim A.17.** *For each color $c \in C$, there exists an orientation $\vec{v} \in \{-1, 1\}^d$ such that $Y_c + B_{\vec{v}} \subseteq [-\frac{1}{2} - \varepsilon, \frac{1}{2} + \varepsilon]^d$.*

---

[12] For every $\vec{y} \in Y_c$ we have (by definition of $Y_c$) that $\gamma(\vec{y}) = c$ and (by definition of $\gamma$) that $\gamma(\vec{y}) = \chi(f(\vec{y}))$ showing that $\chi(f(\vec{y})) = c$ and thus (by definition of $X_c$) that $f(\vec{y}) \in X_c$.

*Proof of Claim.* Let $\vec{v}$ be an orientation given in A.16 for color $c$. We get the following chain of containments:

$$Y_c + B_{\vec{v}} = Y_c + \left( \prod_{i=1}^{d} -v_i \cdot (0, \varepsilon) \right) \qquad \text{(Def'n of } B_{\vec{v}})$$

$$\subseteq \left( \prod_{i=1}^{d} v_i \cdot (-\tfrac{1}{2}, \tfrac{1}{2} + \varepsilon] \right) + \left( \prod_{i=1}^{d} -v_i \cdot (0, \varepsilon) \right) \qquad \text{(A.16)}$$

$$= \left( \prod_{i=1}^{d} v_i \cdot (-\tfrac{1}{2}, \tfrac{1}{2} + \varepsilon] \right) + \left( \prod_{i=1}^{d} v_i \cdot (-\varepsilon, 0) \right) \qquad \text{(Factor a negative)}$$

$$= \prod_{i=1}^{d} v_i \cdot (-\tfrac{1}{2} - \varepsilon, \tfrac{1}{2} + \varepsilon) \qquad \text{(Minkowski sum of rectangles)}$$

$$\subseteq [-\tfrac{1}{2} - \varepsilon, \tfrac{1}{2} + \varepsilon]^d. \qquad (v_i \in \{-1, 1\})$$

This proves the claim. ∎

We also claim that $Y_c + B_{\vec{v}}$ has a substantial measure.

**Subclaim A.18.** *For each color $c \in C$ and any orientation $\vec{v} \in \{-1, 1\}^d$, the set $Y_c + B_{\vec{v}}$ is Borel measurable and $m(Y_c + B_{\vec{v}}) \geq m_{out}(Y_c) \cdot \left( 1 + \frac{\varepsilon}{1+\varepsilon} \right)^d$.*

*Proof of Claim.* Let $M = (1 + \varepsilon)^d$ which is the measure of $\prod_{i=1}^{d} v_i \cdot (-\tfrac{1}{2}, \tfrac{1}{2} + \varepsilon]$, and because by A.16, $Y_c$ is a subset of this set, we have $m_{out}(Y_c) \leq M$.

We have that $Y_c + B_{\vec{v}}$ is Borel measurable and that $m\left( Y_c + B_{\vec{v}} \right) \geq \left( m_{out}(Y_c)^{\frac{1}{d}} + \varepsilon \right)^d$ by A.6 (see details[13]). Thus, we have the following chain of inequalities:

$$m(Y_c + B_{\vec{v}}) \geq \left( m_{out}(Y_c)^{1/d} + \varepsilon \right)^d \qquad \text{(Above)}$$

$$= M \cdot \left( \frac{m_{out}(Y_c)^{1/d}}{M^{1/d}} + \frac{\varepsilon}{M^{1/d}} \right)^d \qquad \text{(Factor out } M)$$

$$\geq M \cdot \left( \frac{m_{out}(Y_c)}{M} \right) \cdot \left( 1 + \frac{\varepsilon}{M^{1/d}} \right)^d \qquad \text{(A.7)}$$

$$= m_{out}(Y_c) \cdot \left( 1 + \frac{\varepsilon}{1 + \varepsilon} \right)^d \qquad \text{(Simplify and use } M = (1 + \varepsilon)^d)$$

∎

Now, consider the indexed family $\mathcal{A} = \left\{ Y_c + B_{\vec{v}(c)} \right\}_{c \in C}$ (where $\vec{v}(c)$ is an orientation for $c$ as in A.16 and A.17) noting that it has finite cardinality because $C$ has finite cardinality. Considering the

---

[13]Note that for the $\ell_\infty$ norm, the measure of the unit ball is $v_{\|\cdot\|_\infty, d} = 2^d$. Then note that $B_{\vec{v}}$ is an open orthant of an $\varepsilon$ ball with respect to $\ell_\infty$, so is in fact itself an $\frac{\varepsilon}{2}$ ball with respect to $\ell_\infty$. This is why we get "$\varepsilon$" instead of the "$2\varepsilon$" in A.6. We could translate this open ball to the origin and translate the set $Y_c$ accordingly to get the same Minkowski sum without changing the measures, and after doing so we could apply A.6 verbatim.

sum of measures of sets in $\mathcal{A}$, we have the following:

$$\sum_{A \in \mathcal{A}} m(A) = \sum_{c \in C} m\left(Y_c + B_{\vec{v}(c)}\right) \qquad \text{(Def'n of } \mathcal{A}\text{; measurability was shown above)}$$

$$\geq \left(1 + \frac{\varepsilon}{1+\varepsilon}\right)^d \cdot \sum_{c \in C} m_{out}(Y_c) \qquad \text{(A.18 and linearity of summation)}$$

$$\geq \left(1 + \frac{\varepsilon}{1+\varepsilon}\right)^d \cdot m_{out}\left(\bigcup_{c \in C} Y_c\right)$$
$$\text{(Countable/finite subaddativity of outer measures)}$$

$$= \left(1 + \frac{\varepsilon}{1+\varepsilon}\right)^d \cdot m_{out}\left(\left[-\tfrac{1}{2} - \varepsilon, \tfrac{1}{2} + \varepsilon\right]^d\right) \quad \text{(The } Y_c\text{'s partition } \left[-\tfrac{1}{2} - \varepsilon, \tfrac{1}{2} + \varepsilon\right]^d\text{)}$$

$$= \left(1 + \frac{\varepsilon}{1+\varepsilon}\right)^d \cdot (1 + 2\varepsilon)^d \qquad \text{(Evaluate outer measure)}$$

By A.17, each member of $\mathcal{A}$ is a subset of $\left[-\tfrac{1}{2} - \varepsilon, \tfrac{1}{2} + \varepsilon\right]^d$, so by 5.1, there exists a point $\vec{p} \in \left[-\tfrac{1}{2} - \varepsilon, \tfrac{1}{2} + \varepsilon\right]^d$ that belongs to at least

$$\left\lceil \frac{\left(1 + \frac{\varepsilon}{1+\varepsilon}\right)^d \cdot (1 + 2\varepsilon)^d}{(1 + 2\varepsilon)^d} \right\rceil = \left\lceil \left(1 + \frac{\varepsilon}{1+\varepsilon}\right)^d \right\rceil$$

sets in $\mathcal{A}$. That is, $\vec{p}$ belongs to $Y_c + B_{\vec{v}(c)}$ for at least $\left\lceil \left(1 + \frac{\varepsilon}{1+\varepsilon}\right)^d \right\rceil$ colors $c \in C$. For each such color $c$, it follows that $\vec{p} + (-\varepsilon, \varepsilon)^d$ intersects $Y_c$ (see justification[14]). Note that with respect to the $\ell_\infty$ norm, $\vec{p} + (-\varepsilon, \varepsilon)^d = B_\infty^\circ(\varepsilon, \vec{p})$ showing that $B_\infty^\circ(\varepsilon, \vec{p})$ contains points of at least $\left\lceil \left(1 + \frac{\varepsilon}{1+\varepsilon}\right)^d \right\rceil$ colors (according to the coloring of $\gamma$ since we are discussing sets $Y_c$).

What we really want, though, is a point in the unit cube that has this property rather than a point in the extended cube, and we want it with respect to the original coloring $\chi$ rather than the extended coloring $\gamma$. We will show that the point $\vec{f}(\vec{p})$ suffices.

**Subclaim A.19.** *If $c \in C$ is a color for which $B_\infty^\circ(\varepsilon, \vec{p}) \cap Y_c \neq \emptyset$, then also $B_\infty^\circ(\varepsilon, \vec{f}(\vec{p})) \cap X_c \neq \emptyset$.*

*Proof of Claim.* Let $\vec{y} \in B_\infty^\circ(\varepsilon, \vec{p}) \cap Y_c$. Then because $\vec{y} \in B_\infty^\circ(\varepsilon, \vec{p})$, we have $\|\vec{y} - \vec{p}\|_\infty < \varepsilon$, so for each coordinate $i \in [d]$, $|y_i - p_i| < \varepsilon$. It is easy to analyze the 9 cases (or 3 by symmetries) arising in the definition of $f$ to see that this implies $|f(y_i) - f(p_i)| < \varepsilon$ as well (i.e. $f$ maps pairs of values in its domain so that they are no farther apart), thus $\|\vec{f}(\vec{y}) - \vec{f}(\vec{p})\|_\infty < \varepsilon$ and thus $\vec{f}(\vec{y}) \in B_\infty^\circ(\varepsilon, \vec{f}(\vec{p}))$.

Also, as justified in a prior footnote[12], for any $\vec{y} \in Y_c$ we have $\vec{f}(\vec{y}) \in X_c$ so that $\vec{f}(\vec{y}) \in B_\infty^\circ(\varepsilon, \vec{f}(\vec{p})) \cap X_c$ which shows that the intersection is non-empty. ∎

Thus, because $B_\infty^\circ(\varepsilon, \vec{p})$ intersects $Y_c$ for at least $\left\lceil \left(1 + \frac{\varepsilon}{1+\varepsilon}\right)^d \right\rceil$ choices of color $c \in C$, by A.19 $\vec{f}(\vec{p})$ is a point in the unit cube which intersects $X_c$ for at least $\left\lceil \left(1 + \frac{\varepsilon}{1+\varepsilon}\right)^d \right\rceil$ different colors $c \in C$. That is, this ball contains points from at least this many of the original color sets.

The final step in the proof of the theorem is to clean up the expression with an inequality. Note that $C$ must contain of at least $2^d$ colors because each of the $2^d$ corners of the unit cube must be assigned a

---

[14]If $\vec{p} \in Y_c + B_{\vec{v}(c)} \subseteq Y_c + (-\varepsilon, \varepsilon)^d$, then by definition of Minkowski sum there exists $\vec{y} \in Y_c$ and $\vec{b} \in (-\varepsilon, \varepsilon)^d$ such that $\vec{p} = \vec{y} + \vec{b}$ so $Y_c \ni \vec{y} = \vec{p} - \vec{b} \in \vec{p} + (-\varepsilon, \varepsilon)^d$ demonstrating that these two sets contain a common point.

unique color since any pair of corners belong to an opposite pair of faces on the cube. For this reason it is trivial that for $\varepsilon > \frac{1}{2}$ there is a point $\vec{p}$ such that $B_\infty^\circ(\varepsilon, \vec{p})$ intersects at least $2^d$ colors: just let $\vec{p}$ be the midpoint of the unit cube. Thus, the only interesting case is $\varepsilon \in (0, \frac{1}{2}]$, and for such $\varepsilon$ we have $1 + \varepsilon \le \frac{3}{2}$ and thus $\frac{\varepsilon}{1+\varepsilon} \ge \frac{2}{3}\varepsilon$ showing that $\left(1 + \frac{\varepsilon}{1+\varepsilon}\right)^d \ge (1 + \frac{2}{3}\varepsilon)^d$. This completes the proof of the theorem. □

# B   Measure Theory

Throughout this section, we use the word "countable" to mean finite or countably infinite.

**Fact B.1** ($\ell_\infty$ Diameter Ball). *Let $d \in \mathbb{N}$ and $X \subseteq \mathbb{R}^d$ be a bounded set with diameter $D$ (with respect to $\ell_\infty$). Then there exists $\vec{p} \in \mathbb{R}^d$ such that $X \subseteq \overline{B}_{D/2}(\vec{p})$. As a consequence, $m_{out}(X) \le D^d$ where $m_{out}$ denotes outer Lebesgue measure.*

*Proof Sketch.* For each coordinate $i \in [d]$, consider the set $X_i = \{\pi_i(\vec{x}): \vec{x} \in X\} \subseteq \mathbb{R}$ of the $i$th coordinates of each point in $X$. The infimum and supremum are distance at most $D$ apart, because otherwise there would be points $\vec{y}, \vec{z} \in X$ such that $|\pi_i(\vec{z}) - \pi_i(\vec{y})| > D$ which means $\|\vec{z} - \vec{y}\|_\infty > D$. Thus taking $\vec{p} = \langle \frac{\inf(X_i) + \sup(X_i)}{2} \rangle_{i=1}^d$ we have $X \subseteq \prod_{i=1}^d [\inf(X_i), \sup(X_i)] \subseteq \vec{p} + [-\frac{D}{2}, \frac{D}{2}]^d = \overline{B}(D/2, \vec{p})$. □

**Fact B.2.** *If $\mu$ is a measure and $\mathcal{A}$ is a (possibly uncountable) family of pairwise disjoint measurable sets, then*

$$\mu\left(\bigsqcup_{A \in \mathcal{A}} A\right) \ge \sum_{A \in \mathcal{A}} \mu(A).$$

*Proof.* By definition of the arbitrary summation (c.f. [20, p. 11]) we have

$$\sum_{A \in \mathcal{A}} \mu(A) \stackrel{\text{def}}{=} \sup\left\{\sum_{A \in \mathcal{F}} \mu(A) : \mathcal{F} \subseteq \mathcal{A}, \ \mathcal{F} \text{ finite}\right\}$$

and for any $\mathcal{F} \subseteq \mathcal{A}$ we have

$$\mu\left(\bigsqcup_{A \in \mathcal{A}} A\right) \ge \mu\left(\bigsqcup_{A \in \mathcal{F}} A\right) = \sum_{A \in \mathcal{F}} \mu(A).$$

Thus $\mu(\bigsqcup_{A \in \mathcal{A}} A)$ is an upper bound for the set $\left\{\sum_{A \in \mathcal{F}} \mu(A) : \mathcal{F} \subseteq \mathcal{A}, \ \mathcal{F} \text{ finite}\right\}$ and thus greater than or equal to the supremum. □

**Fact B.3** (Interchange of Countable Sums with Non-negative Terms). *If $I, J$ are countable sets, and $a_{i,j} \ge 0$ for all $(i, j) \in I \times J$, then*

$$\sum_{i \in I} \sum_{j \in J} a_{i,j} = \sum_{j \in J} \sum_{i \in I} a_{i,j}$$

*Proof.* This can be proved directly via basic analysis methods if $I$ and $J$ are assumed to be $\mathbb{N}$ and the definition of the infinite sum as a limit of finite sums is used. Alternatively, viewing the summation as an integral over a countable measure space, this can be viewed as a corollary to Tonelli's theorem. □

**Lemma B.4** (Upper Bound Measure of Multiplicity). *Let $n \in \mathbb{N}$. Let $X$ be a measurable set in some measure space (the measure being denoted by $\mu$) and let $\mathcal{A}$ be a countable family of measurable subsets of $X$ such that for each $x \in X$, $x$ belongs to at most $n$ members of $\mathcal{A}$. Then*

$$\sum_{A \in \mathcal{A}} \mu(A) \le n \cdot \mu(X).$$

*Proof.* [15] For ease of indexing, assume $\mathcal{A} = \{A_i\}_{i=1}^\infty$. Let $I_{A_i}$ denote the indicator function for $A_i$, and note that by hypothesis, for any $k \in \mathbb{N}$, the function $\sum_{i=1}^k I_{A_i}$ is bounded above by the constant

---

[15]We thank an anonymous researcher who pointed out that our original proof was overcomplicating things and pointing us to this elementary proof.

function $n$. Thus, recalling that $\mu(A_i) = \int_X I_{A_i} d\mu$ (because $A_i \subseteq X$) we have the following for any $k \in \mathbb{N}$:

$$\sum_{i=1}^{k} \mu(A_i) = \sum_{i=1}^{k} \left[ \int_X I_{A_i} d\mu \right] = \int_X \left[ \sum_{i=1}^{k} I_{A_i} \right] d\mu \leq \int_X n \, d\mu = n \cdot \mu(X).$$

Since this holds for all $k$, it also holds in the limit:

$$\sum_{A \in \mathcal{A}} \mu(A) = \sum_{i=1}^{\infty} \mu(A_i) = \lim_{k \to \infty} \left[ \sum_{i=1}^{k} \mu(A_i) \right] \leq n \cdot \mu(X).$$

$\square$

**Corollary B.5** (Lower Bound Cover Number). *Let $X$ be a measurable set in some measure space (the measure being denoted by $\mu$) such that $0 < \mu(X) < \infty$. Let $\mathcal{A}$ be a countable family of measurable subsets of $X$ such that $\sum_{A \in \mathcal{A}} \mu(A) < \infty$. Then there exists $x \in X$ such that $x$ belongs to at least $\left\lceil \frac{\sum_{A \in \mathcal{A}} \mu(A)}{\mu(X)} \right\rceil$-many members of $\mathcal{A}$.*

*Proof.* First observe that by hypothesis, $\left\lceil \frac{\sum_{A \in \mathcal{A}} \mu(A)}{\mu(X)} \right\rceil$ is finite. Suppose for contradiction that each $x \in X$ belongs to strictly less than $\left\lceil \frac{\sum_{A \in \mathcal{A}} \mu(A)}{\mu(X)} \right\rceil$-many members of $\mathcal{A}$. Let $n = \left\lceil \frac{\sum_{A \in \mathcal{A}} \mu(A)}{\mu(X)} \right\rceil - 1$ (noting that $n < \frac{\sum_{A \in \mathcal{A}} \mu(A)}{\mu(X)}$). Then each $x \in X$ belongs to at most $n$-many members of $A$, so we have

$$\sum_{A \in \mathcal{A}} \mu(A) \leq n \cdot \mu(X) \qquad \text{(Lemma B.4)}$$

$$< \frac{\sum_{A \in \mathcal{A}} \mu(A)}{\mu(X)} \mu(X) \qquad \left(0 < \mu(X) < \infty \text{ and } n < \frac{\sum_{A \in \mathcal{A}} \mu(A)}{\mu(X)}\right)$$

$$= \sum_{A \in \mathcal{A}} \mu(A)$$

which is a contradiction. $\square$

**Remark B.6.** *In Corollary B.5 above, it was important that we required $\sum_{A \in \mathcal{A}} \mu(A)$ to be finite. If we allowed it to be infinite, then the claim would have been that there was some $x \in X$ belonging to infinitely many members of $\mathcal{A}$, but this is in general not true (see B.7 below). Nonetheless, it is true (and a straightforward corollary of the above) that if $\sum_{A \in \mathcal{A}} \mu(A) = \infty$, then for any $n \in \mathbb{N}_0$, there exists a point $x_n \in X$ that is contained in at least $n$-many sets of $\mathcal{A}$. The distinction is that this point might have to depend on the choice of $n$.*

**Example B.7** (Harmonic Cover of Open Unit Interval). *Let $X = (0, 1)$ be equipped with the Borel or Lebesgue measure $\mu$. Let $\mathcal{A} = \left\{ (0, \frac{1}{i}) : i \in \mathbb{N} \right\}$. Then $\sum_{A \in \mathcal{A}} \mu(A) = \sum_{i \in \mathbb{N}} \frac{1}{i} = \infty$. For any $n \in \mathbb{N}$, we can consider the point $x_n = \frac{1}{n+1}$ which is contained in $(0, \frac{1}{i})$ for $i \in [n]$ and not for any other $i$, so it belongs to exactly $n$ sets in $\mathcal{A}$.*

*However, no point in $X$ belongs to infinitely many sets in $\mathcal{A}$. To see this, consider an arbitrary point $x \in X = (0, 1)$. Then for sufficiently large $i \in \mathbb{N}$, $x \notin (0, \frac{1}{i})$ so $x$ belongs to only finitely many members of $\mathcal{A}$.*

The prior results have been stated in typical measure theory notation, but in the body of the paper we present B.5 as follows for $\mathbb{R}^d$ specifically with notation matching what is used elsewhere in the paper.

Proof of Proposition A.1 This follows trivially from B.5 and B.6.

