# OpenReview forum: "Replicability in Learning: Geometric Partitions and KKM-Sperner Lemma"
_NeurIPS.cc/2024/Conference — NeurIPS 2024 poster_

### Official Review · Reviewer_1env · 2024-06-27

**Soundness:** 4
**Presentation:** 4
**Contribution:** 3
**Rating:** 6
**Confidence:** 4

**Summary:**

The authors study the list-replicable coin problem and a closely related underlying geometric problem of constructing ($k,\varepsilon$)-secluded partitions of $\mathbb{R}^d$. The authors resolve the optimal trade-off between $k,\varepsilon$, and $d$ in the latter, and as a corollary give a new set of upper bounds for the list-replicable coin problem trading off list-size and sample complexity for the former.

The coin problem asks, given coins with biases $p_1,\ldots , p_d$, estimate the biases of each coin up to accuracy $\nu$. An algorithm for the coin problem is called *L-list-replicable* if with high probability over samples, it’s output (for any fixed set of biases) is one of $L$ possible ($\nu$-correct) bias vectors. List replicability is closely related to other core notions of stability in learning such as differential privacy, global stability, and replicability.

Prior work established that it is possible the list-replicable solve the coin problem with list size $d+1$ and $d^2$ samples per coin. The former parameter, the list size, was known to be optimal, but it was open whether the number of samples could be improved. The main "learning" result of this paper is an upper and conditional lower bound for the list-replicable coin problem giving a trade-off between list size and samples. The upper bound states for any list size $L \in [2^d] \setminus [d]$, there is an $L$-list-replicable algorithm using $\tilde{O}(\frac{d^2}{\nu^2\log^2(L)})$ samples per coin. The lower bound states that this trade-off is tight for any algorithm based on `secluded partitions’.

The true main result of the paper is a near-tight bound on the construction of ($k,\varepsilon$)-secluded partitions of $\mathbb{R}^d$. A ($k,\varepsilon$)-secluded partition is a partition of $\mathbb{R}^d$ such that for any point $p \in \mathbb{R}^d$, the $\ell_\infty$ $\varepsilon$-ball around $p$ touches at most $k$ members of the partition. It is known that any  ($k,\varepsilon$)-secluded partition leads to a $k$-list replicable algorithm with roughly $1/\varepsilon^2$ blow-up in samples per coin over the non-list-replicable coin problem.

Prior works established the existence of $(d+1,\frac{1}{2d})$-secluded partitions. The authors roughly show for any $\varepsilon$ one can construct a $((1+2\varepsilon)^d,\varepsilon)$-secluded partition and that this is tight.

The authors also establish a related "neighborhood" variant of the KKM/Sperner Lemma, stating that any coloring of [0,1]^d where no color touches opposite faces must have a point $p$ such that the $\ell_\infty-\varepsilon$-ball around $p$ touches $(1+O(\varepsilon))^d$ many colors. This is in some sense a `quantified/robust’ version of KKM, which states there is a point where every $\varepsilon$-ball sees at least $d+1$ colors, though it is actually incomparable.

**Strengths:**

Algorithmic stability is a natural and recently fruitful topic in learning theory, closely related to core notions including differential privacy. Understanding the `cost’ trade-off between samples and strength of stability (in this case, list size) is a very natural problem.

The coin problem is a fundamental learning problem that has also shown up as a subroutine in analysis of replicable tasks, e.g. reinforcement learning in ("Replicability in Reinforcement Learning") and hypothesis testing in ("Replicability in High Dimensional Statistics"). The authors prove a new upper bound for this problem that interpolates between (essentially) no overhead at the `trivial’ list size of $2^d$, to $d^2$ overhead at the optimal list size $d+1$. The authors show this is tight assuming the use of secluded partitions.

The problem of secluded partitions itself, while formally a purely geometric result, is a natural problem and does have a similar flavor to recent techniques used for lower bounds in algorithmic stability. I would not be surprised if the author’s new neighborhood lemma sees future use in learning.

Finally, the paper (or at least the introduction) is very well-written, with good proof intuition clearly laid out in the introduction and a good overview of prior literature.

**Weaknesses:**

This work’s only substantial weakness lies in its scope for NeurIPS: while the geometric results proved are indeed fundamental, the resulting implications within replicable learning are, at the moment, a bit lack-luster.

In particular, while the upper bound result itself mentioned above is nice, it is fairly close to being a corollary of the previous work “List and Certificate Complexities in Replicable Learning". All that is needed over the original work is the new secluded partition construction, which is just a product of the construction used in the original paper. Based on the definition itself (namely working in $\ell_\infty$), it is elementary to see such a construction will work, and while not formally stated in the original paper (which just focused on achieving $d+1$ list complexity), it was already pretty clear this was possible.

Thus in terms of novelty, the core contribution of this paper is really the *lower bound* on secluded partitions. This is a beautiful result geometrically, but because we do not know secluded partitions are necessary for list-replicable learning, it does not actually give a lower bound on the coin problem, nor do the authors yet have another use for the bound or related neighborhood KKM Lemma. I believe these notions will indeed prove useful, which is why I am still recommending this paper be accepted, but at the moment the learning results are a bit too weak for a stronger recommendation.

**Questions:**

N/A

**Limitations:**

The authors have appropriately discussed the limitations of their work.

---

> ### Author Rebuttal · Authors · 2024-08-07
>
> Response to Weaknesses Comment:
>
> The concern regarding the scope to NeurIPS is addressed in the General Response as well as the response to Reviewer W1GL. We agree the main novelty and technicality is in establishing the lower bound. However, the upper bound is critical to complete the picture. Thank you for the optimistic remark that these notions have the potential to be useful in the future.

---

> > ### Comment · Reviewer_1env · 2024-08-07
> > **Rebuttal Acknowledgement**
> >
> > We acknowledge the authors' rebuttal. We agree with the authors that the material arose in the context of replicability, has some implications therein, and may be of use in algorithmic stability in the future. I think the results are important, and will be of interest to a specialized subset of the NeurIPS community (including, e.g., myself). Nevertheless, my general opinion as stated in the review stands: the paper should be accepted, but is held back from a stronger recommendation by its scope with respect to learning.

---

> > > ### Author Response · Authors · 2024-08-12
> > >
> > > Thank you for reviewing the rebuttal and your positive response.

---

### Official Review · Reviewer_D2Ky · 2024-07-07

**Soundness:** 4
**Presentation:** 4
**Contribution:** 4
**Rating:** 7
**Confidence:** 4

**Summary:**

This paper studies connections between (i) list-replicability, a well-studied relaxation of the standard replicable learning framework and (ii) the design of partitions of the Euclidean space, in particular $(k,\epsilon)$-secluded partitions. A partition $P$ will be called $(k, \epsilon)$-secluded if for any $d$-dimensional vector $x$, a box centered at $x$ with side-length $2\epsilon$ intersects with at most $k$ members of  $P$. Crucially, the lis size of a list replicable learner is closely related to $k$.

The main results of the paper are near-optimal trade-offs between $k$ and $\epsilon$ for secluded partitions. These imply some limitations on the design of list-replicable algorithms based on such partitions of the space.

**Strengths:**

Since there is a $(k, \epsilon) = (d+1, 1/(2d))$-secluded partition of the $d$-dimensional Euclidean space, one natural question is whether one can improve on $k$ or $\epsilon$. For $k$, we knew that this is tight. The main result of the paper states that essentially the dependence on $\epsilon$ is also tight: for any $k \leq 2^d$, it must be the case that $\epsilon \in \tilde{O}(1/d)$. Hence, even by relaxing the number of intersections to be $k = \mathrm{poly}(d),$ one cannot get $\epsilon = \omega(1/d)$. I find this result interesting since it raises new questions on how to design improved list-replicable learners.

The second result of the paper is a construction of a $(k, \epsilon)$-secluded partition for (roughly) any $k \in [2^d]$. Each choice of $k$ gives a related value for $\epsilon$.

I find the connections raised in this paper quite nice. I feel that the geometric insight is very useful and could inspire future work in the area of replicable algorithms. The results of the paper are clear and I enjoyed reading the paper. Finally, I think that the paper is technically interesting and the arguments are nice.

**Weaknesses:**

Some suggestions in terms of presentation:

1) Between line 91-95, I would add some more discussion about Sperner's lemma and the terminology used.

**Questions:**

1) Do you have any intuition on what would happen if we replaced the $L_\infty$ norm with some other norm in terms of list-replicability?

2) There is a series of works that establish connections between replicability and approximate DP. Is there some similar connection to list-replicability?

3) Returning to the geometric connections to list replicability, is there some similar geometric intuition for DP?

**Limitations:**

No limitations.

---

> ### Author Rebuttal · Authors · 2024-08-07
>
> Response to Suggestion:
>
> Thank you for the suggestion. We will add the discussion along the lines you propose.
>
> Response to Questions:
>
> Q1: Unit ball in $\ell_p$ norm is a subset of a unit ball in $\ell_\infty$ norm. Thus secluded partitions with respect to $\ell_\infty$ norm are also secluded with respect to $\ell_p$ norm. Based on this, we can get a $d+1$-list replicable algorithm for the d-biased coin problem (where the approximation is in $\ell_p$ norm), using the known relation between the secluded partition and list replicable algorithms. In the present work, we also establish a generic version of Theorem 3.1 that applies to other norms. This result appears in the appendix as Theorem A.8. From this theorem, for example for $\ell_2$ norm, we obtain that $k > (1+\varepsilon\sqrt(2\pi e/d))^d$.
>
> Q2 and Q3: We know from the work reported in [11], the global stability parameter is inverse to the list size. Prior work established the relationship between DP and stability [9]. In particular, from these works it follows that  for a concept class $\cal C$ a learning algorithm with a polynomial list size with polynomially many samples implies that the concept class is DP learnable with polynomial sample complexity.  Admittedly, we are not experts on DP, and our intuition there is limited.
>
> We thank the reviewer for very encouraging remarks about the usefulness of our work in the future.

---

### Official Review · Reviewer_W1GL · 2024-07-14

**Soundness:** 3
**Presentation:** 2
**Contribution:** 2
**Rating:** 5
**Confidence:** 4

**Summary:**

This work studies secluded partitions, which appear in the context of list-replicable learning (among other geometric/TCS applications). The complement known bounds on list complexity, the authors present new upper-bounds on the tolerance parameter as a function of the list complexity k and the ambient dimension d. They show a construction of a secluded partition that roughly achieves these bounds, showing the optimality of their bounds. The secluded partition results are then applied to give a "neighborhood" version of Sperner's lemma, for $\ell_\infty$ neighborhoods.

**Strengths:**

This work provides novel results with potentially broad applications in geometry and computer science. Sperner's lemma is widely useful across disciplines, and this neighborhood variant may prove similarly broadly useful.

**Weaknesses:**

As written, I don’t think this work is a good fit for NeurIPS. This work  studies secluded partitions, which have connections to list-replicable algorithms for, e.g., the d-coin bias estimation problem as mentioned by the authors. In this sense the results have applications to replicable learning, but the actual application of the secluded partitions results give somewhat marginal replicable learning results. Theorem 3.1 essentially shows that one cannot meaningfully tradeoff list complexity for better secluded partition tolerance, and therefore cannot hope to improve sample complexity for the d-coin bias problem via the secluded partition approach, but says nothing of other approaches or other problems.

If I try to read this as a learning paper where the focus is improving sample complexity of list-replicable d-coin bias estimation algorithms, it seems like there should be more attention paid in related work to replicable (in the sense of ILPS22) algorithms for the d-coin bias estimation problem as well, as this has been studied in [1], [2], and recently (posted after NeurIPS submission deadline) [3].

[1] “Stability is Stable” Bun, Gaboardi, Hopkins, Impagliazzo, Lei, Pitassi, Sivakumar, Sorrell
[2] “Replicability in Reinforcement Learning” Karbasi, Velegkas, Yang, Zhou ’23
[3] “Replicability in High Dimensional Statistics” Hopkins, Impagliazzo, Kane, Liu, Ye ‘24

Typos/suggested edits:

Abstract

“We show that for each d” -> “we show for each d”

Section 1 pg 2
Brower -> Brouwer
should be seen as a fundamental endeavor. .

Section 2 page 2
“belongs to a list L consisting of almost k good hypotheses”

“over which a family of distribution” -> “over which a family of distributions”

page 3

“the partitions considered in this work have a non-zero” -> “the partitions considered in this work have non-zero

Theorem 2.4 could be written a little more clearly. It would be good to define the mapping implicit in the term “coloring” and clarify that opposite faces mean faces of the hypercube

Section 3

“in general a $(k, \varepsilon)$-secluded partitions”

“Can we improve this and construct a (d+1, \omega(1/d)-secluded partition?”

“is the following result upper bound result”

“Till this work we know”

“There exist a k-list replicable algorithm”

“Spener/KKM Lemma”

Section 4
“A learning algorithm A to be $(n, \eta)$-globally stable”

Section 5.1

Need a period before Thus/ what we do is to “replace”

I found the proof sketch for Theorem 3.1 had a few seemingly unnecessary detours that were a bit distracting. For instance, deriving the lower bound of 1 before deriving the desired bound. The footnote could be more precise (what does “becomes the wrong inequality” mean?)

Need a period before “Now that we have dealt with both issues” on page 6.

“So because there is a ceiling involved” could be made more precise, as could “by our change of perspective.”

Section 5.2

There’s a $d_n$ that should be $d_i$ in Definition 5.2

Section 6

“We also constructed secluded partitions for a wide all $k$”

The second paragraph of Section 6 is very vague, and doesn’t mention that the similar upper bounds are for general lp norms.

**Questions:**

What direct implications do these results have for list-replicable learning?

How do these implications compare to what is known for $\rho$-replicable learning for similar problems?

**Limitations:**

Yes, the authors have addressed all limitations.

---

> ### Author Rebuttal · Authors · 2024-08-07
>
> Response to Weaknesses Comment:
>
> This weakness regarding the scope is addressed in the general response, and we reiterate it here. Our work is motivated by the connection of geometric/topological tools to list replicability (secluded partitions and Sperner/KKM Lemmas). Driven by this connection, the current work undertakes an in-depth investigation of secluded partitions, a tool used to establish list replicability results. We believe that understanding the power and limitations of tools employed in replicability research is a significant contribution to the field and falls within the scope of NeurIPS.
>
> Our investigation into secluded partitions not only led us to a comprehensive understanding of secluded partitions but also to a new neighborhood version of Sperner/KKM lemma. Sperner/KKM lemma and its variants (such as Poincare-Miranda) have proven to be critical tools in replicable learning and have applications in broader areas of computer science. We believe that this new version neighborhood version of Sperner/KKM lemma will have significant applications in learning. Other reviewers (D2ky, 1env) have expressed similar sentiments.
>
> Regarding related works, thank you for suggested references. We will make the related work section more comprehensive in the final version taking into consideration all the references, including the ones that appeared after NeurIPS deadline.
>
> Thank you for carefully reading and pointing out the typos and suggestions for improvements. We will fix the typos and will incorporate your suggestions in the final version.
>
> Response to Questions:
>
> Q1: Our secluded partition construction (Theorem 3.2) leads to corollary 3.3 which gives a tradeoff between list size and sample complexities. For example, if we allow the list size to be $2^{\sqrt{d}}$ then we can get sample complexity of $\tilde O(d/\nu^2)$ per coin, which is a new result. Theorem 3.1 shows the limitations of secluded partitions as a tool.
>
> Q2: It is possible to go from list-replicable algorithms to $\rho$-replicable algorithms (using ideas from Goldreich [25]).  In $\rho$-replicability,  one is allowed to choose a random string. Once a random string is fixed, the rest of the computation is perfectly replicable (i.e.,  with list size 1). Goldreich’s construction implies that a $\ell$-list replicable algorithm leads to a $\rho$-replicable algorithm where the length of the random string is of the order of  $O(\log l)$ (with additional dependency on  $\rho^2$). However, such a transformation leads to a sample complexity blow-up. This is already observed in [16].

---

> > ### Comment · Reviewer_W1GL · 2024-08-12
> >
> > Thank you to the authors for addressing my concerns regarding fit and for committing to expanding the related work section. I still believe that this paper would be better-appreciated in another venue, but given that I enjoyed the work and the overall positive scores of other reviewers, it seems like I might be wrong and there will be enough of an interested audience at NeurIPS that I will revise my score.

---

> > > ### Author Response · Authors · 2024-08-12
> > >
> > > Thank you for reviewing the rebuttal and your response and being positive about the work.

---

### Official Review · Reviewer_Srve · 2024-07-16

**Soundness:** 4
**Presentation:** 3
**Contribution:** 2
**Rating:** 5
**Confidence:** 4

**Summary:**

In this work the authors study various connections between geometric constructions in $R^d$, in particular various $(k,\varepsilon)$-secluded partitions, and its applications to list-replicable learnability. This connection was originally observed in prior work, and the authors provide stronger quantitative results. More specifically, their first main results shows an upper bound on the $\varepsilon$ for every (non-trivial) choice of the parameter $k$ of the secluded partition. This implies a lower bound on the number of samples needed to design a $k$-list-replicable algorithm for the problem of coin estimation through secluded partitions, and can be interpreted as a negative result: if $k = poly(d)$, then one needs $d^2$ samples per coin to design a $k$-list-replicable algorithm (through secluded partitions). This improves upon prior work which shows an upper bound on $\varepsilon$ only for $k = d+1$
Their next main provides a construction of $(k,\varepsilon)$-secluded partitions for all the (non-trivial) values of $k$ that is optimal w.r.t. $\varepsilon$ up to polylogarithmic factors. Due to the previous connection, this implies the existence of $k$-list-replicable algorithms whose sample complexity is determined by $\varepsilon$.
Lastly, the authors use the techniques they have developed to prove a "neighborhood" version of the cubical Sperner's/KKM lemma.

**Strengths:**

-The constructions that the authors present are mathematically elegant.

-The paper is written clearly and is easy to follow.

-Their applications to list-replicable learning, which has gained some attention recently, are spelled out in the paper.

-Some of the results, like the extension of Sperner's lemma, can have further applications.

**Weaknesses:**

I am a bit worried that the results are a bit incremental. In particular, the connection between secluded partitions and list-replicability was already observed in prior work. Moreover, I view the result regarding secluded partitions as a negative result, in the sense that in order to get reasonable list complexity through secluded partitions, one needs a significant blow-up in the sample complexity, and this cannot be improved using this approach. Had the lower bound on the sample complexity been against all algorithms (or at least a broader class) I would have found the result more interesting. Similarly, the extension of cubical Sperner's lemma to its neighborhood variant is interesting on its own, but if the main focus of the paper is list-replicability, I am not too sure how much it adds to the story.

Some minor points/typos:
-Line 53: double full-stop.
-Line 56: "the replicable algorithm" -> replicable algorithms.
-Theorem 2.4: it would be better for a broader audience to define face/closure.
-Line 109: the degree parameter -> it.
-Line 339: know -> known.
-In many places: lowerbound -> lower bound
-Line 156: global stability.
-Line 172: extend -> extended.
-Line 112: a ) is missing.
-Line 114: drop result.

**Questions:**

-Do you have a conjecture about the lower bound on the sample complexity of $k$-list replicable learning for algorithms that are not based on secluded partitions?

-Can you explain a bit more about the adaptation of Theorem 5.1 you talk about in Line 231?

**Limitations:**

Adequately addressed.

---

> ### Author Rebuttal · Authors · 2024-08-07
>
> Response to the Weaknesses Comment:
>
> We agree with the reviewer that the connection between secluded partitions and list-replicability was already observed. Indeed, that is our  main motivation to further investigate the possibility of constructing secluded partitions better than those previously known. However, our result is negative and shows that the known constructions cannot be improved by much. We believe negative results are  a significant contribution in the sense that they guide the research directions. Moreover, the techniques used to establish the negative result led to the discovery of the neighborhood-Sperner/KKM lemma. We disagree with the reviewer that the results are incremental because our results provide a comprehensive picture of a secluded partition (both constructions and impossibility results), and for establishing the lower bounds we need quite different techniques (from measure theory and geometry) than those used in the literature.
>
> Thank you for pointing out the typos. We will fix them in the final version.
>
> Response to Questions:
>
> Answer to Question:
>
> Q1. We conjecture that any $(d+1)$-list replicable algorithms for the d-bias coin problem requires $\Omega(d^2/\nu^2)$ samples per coin.
>
> Q2. Adaptation of Theorem 5.1: This is only a slight technicality: Our lower bound theorem works for partitions whose members have bounded outer measure (need not be measurable). However, for Theorem 5.1 to be applied, we need sets $A, B$, and $A+B$ to be measurable. So an adaptation of 5.1 is needed and is given as Lemma A.6, which allows us to work with outer measures.

---

> > ### Comment · Reviewer_Srve · 2024-08-09
> >
> > I would like to thank the authors for their thorough response. Just a clarification on my comment; I didn't imply that the technical contribution of the lower bound is incremental, I do think it's an elegant construction. I meant to say that the result on its own is a bit incremental, since it implies a hardness only for a particular technique/approach for constructing list-replicable algorithms. I also agree that the new variant of the Sperner/KKM lemma is interesting, it just feels a bit disconnected from the main message/story of the rest of the paper.
> >
> > I agree with some of the points the other reviewers have mentioned, but I am still on the positive side about the paper.

---

> ### Author Response · Authors · 2024-08-12
>
> Thank you for reviewing the rebuttal and your positive response.

---

### Author Rebuttal · Authors · 2024-08-07

General Response:

We sincerely thank all the reviewers for their careful reading and thoughtful comments and suggestions.

A common concern raised by the reviewers is the scope and fit of the present work for NeurIPS, which we address here. Our work is motivated by the fundamental connections between geometry/topology and the fast-emerging topic of replicable learning. This connection continues to be revealed. For instance, the works reported in [29, 16] use geometric partitions to design replicable algorithms, while those in [16,12,11] utilize Sperner-KKM-like fixed point theorems from geometry/topology to establish lower bounds (references as appear in the submission). Therefore, it is natural and fundamental to further investigate geometric partitions and Sperner/KKM-type theorems in relation to replicability. We believe that undertaking such investigations is a significant endeavor and should be of interest to the NeurIPS community.

Our work is a comprehensive study of geometric partitions, specifically secluded partitions, which have direct applications to list replicability [16]. Additionally, the work establishes a neighborhood variant of the Sperner/KKM lemma. As  majority of reviewers have pointed out, this neighborhood version is a fundamental contribution with strong potential for applications in replicability research.

Thus, while our motivation to investigate the partition problem is inspired by replicability, our study focuses on geometric partitions and Sperner/KKM-type results. We believe such a study is within the scope of the conference.

---

### Decision · Program_Chairs · 2024-09-25

**Decision:**

Accept (poster)

**Comment:**

The reviewers enjoyed reading the paper and appreciated its content. The only concern raised was that the main technical results seem somewhat loosely connected to the computational notion of replicability. Despite this, the reviewers found the results both interesting and inspiring.